# Combining mass spectrometry and machine learning to discover bioactive peptides

Christian T. Madsen [1] ✉, Jan C. Refsgaard [1,2], Felix G. Teufel [1], Sonny K. Kjærulff[1,2], Zhe Wang[3], Guangjun Meng[3,4], Carsten Jessen [1], Petteri Heljo[1], Qunfeng Jiang[3,5], Xin Zhao[3], Bo Wu[3,6], Xueping Zhou [3,7], Yang Tang[3,8], Jacob F. Jeppesen[1], Christian D. Kelstrup[1], Stephen T. Buckley[1], Søren Tullin[1,9], Jan Nygaard-Jensen[1,9], Xiaoli Chen[3], Fang Zhang[3,10], Jesper V. Olsen [11], Dan Han[3], Mads Grønborg[1,13] & Ulrik de Lichtenberg [1,12,13]

Peptides play important roles in regulating biological processes and form the basis of a multiplicity of therapeutic drugs. To date, only about 300 peptides in human have confirmed bioactivity, although tens of thousands have been reported in the literature. The majority of these are inactive degradation products of endogenous proteins and peptides, presenting a needle-in-a-haystack problem of identifying the most promising candidate peptides from large-scale peptidomics experiments to test for bioactivity. To address this challenge, we conducted a comprehensive analysis of the mammalian peptidome across seven tissues in four different mouse strains and used the data to train a machine learning model that predicts hundreds of peptide candidates based on patterns in the mass spectrometry data. We provide in silico validation examples and experimental confirmation of bioactivity for two peptides, demonstrating the utility of this resource for discovering lead peptides for further characterization and therapeutic development.

Peptides are known to be potent regulators of a diverse set of biological functions and form the basis of around 80 marketed drugs, with an additional 150 peptide-based candidates in clinical development[1]. Examples of well characterized bioactive peptides include GLP-1, Islet amyloid polypeptide (amylin) and Peptide YY (PYY) which regulate glucose stimulated insulin secretion, glycogen deposition in muscle, and colonic mobility, respectively[2–4].

Most bioactive peptides are formed by specific enzymatic cleavages of protein precursors, exemplified by pro-glucagon which is processed, in a tissue-specific manner, into at least nine different peptides with divergent functionalities[5]. Peptide activity may furthermore be modulated by post-translational modifications or extracellular peptidases[6–8]. For most of the well described peptide hormones, the precursor travels via the canonical secretory pathway where it is cleaved by prohormone convertases, trimmed by carboxypeptidase E[9,10] and in some cases amidated by the PAM enzyme (requiring a Glycine in the +1 position). However, not all bioactive peptides follow this path[11–14] (Supplementary Fig. 1). For instance, the antimicrobial peptide Buforin is proteolytically generated from histone H2A[15], illustrating that peptides can be formed even from

---

[1]Global Research Technologies, Novo Nordisk A/S, Maaloev, Denmark. [2]Intomics, Kongens Lyngby, Denmark. [3]Novo Nordisk Research Centre China, Beijing, China. [4]Pulmongene LTD. Rm 502, Building 2, No. 9, Yike Road, Zhongguancun Life Science Park, Changping District, Beijing, China. [5]Innovent Biologics, Inc. DongPing Jie 168, Suzhou, China. [6]QL Biopharmaceutical, Rm 101, Building 7, 20 Life Science Park Road, Beijing, China. [7]Crinetics pharmaceuticals, 10222 Barnes Canyon Rd Building 2, San Diego, CA 92121, USA. [8]Roche R&D Center (China) Ltd, Building 5, 371 Lishizhen Road, 201203 Pudong, Shanghai, China. [9]Boehringer Ingelheim GmbH & Co. KG, Birkendorfer Str. 65, 88397 Biberach, Germany. [10]Structure Therapeutics. 701 Gateway Blvd., South San Francisco, CA 94080, USA. [11]Department of Proteomics, The Novo Nordisk Foundation Center for Protein Research, University of Copenhagen, Copenhagen, Denmark. [12]The Novo Nordisk Foundation, Tuborg Havnevej 19, 2900 Hellerup, Denmark. [13]These authors contributed equally: Mads Grønborg, Ulrik de Lichtenberg. ✉e-mail: ctom@novonordisk.com

intracellular proteins with a function far removed from that of the peptide. Other peptides originate from cleavage of longer peptides, as illustrated by the vasoconstrictive peptide Angiotensin-II which is formed in the blood from the inactive Angiotensin-I peptide by the Angiotensin-converting enzyme (ACE)[16]. ACE also cleaves the vasodilator peptide Bradykinin into an inactive form[17] and is the target of hypertension drugs due to its role in regulating these potent bioactive peptides. When studying biological samples and whole tissues, one should thus expect to observe multiple variants of the same peptides, only some of which are bioactive, as well as differences in abundance, processing, secretion, and degradation across tissues.

Advances in high-throughput omics technologies have over the past decades shed light on the many layers of cellular regulation, and led to the systematic mapping of the human genome, transcriptome, and proteome, as well as the regulatory roles of RNA, epigenetic modifications, and post-translational modifications. In comparison, discovering bioactive peptides and uncovering their mode of action remains a major challenge and to date only about 300 peptides have been characterized in human. The majority of these functionally validated peptides, which include neuropeptides, endocrine peptides, or antimicrobial peptides, have been discovered via years of focused experimentation, rather than by large-scale comprehensive mapping. The fact that peptides are only formed and released in certain tissues and cell types under certain conditions—which may be unknown or difficult to reproduce experimentally—adds to the complexity of identifying bioactive peptides.

In recent years, mass spectrometry (MS) based peptidomics analysis has emerged as a powerful and sensitive tool for systematically mapping the peptide space (the peptidome) present in tissues or biofluids[18–23], including known bioactive peptides, longer inactive precursor peptides, shorter inactive forms, as well as the thousands of inactive degradation products formed as a consequence of natural protein and peptide turnover. MS-based peptidomics analysis is thus well-suited for monitoring the abundances of already known peptides in tissues and cellular extracts but using it as a tool for discovering novel bioactive peptides presents a major needle-in-a-haystack challenge of separating the small set of real bioactive peptides from the vast background of observed degradation products and inactive precursors. The sum of unique peptide sequences reported from small- and large-scale peptidomics experiments during the past decade far exceeds the capacity for functional testing. Methods for identifying and prioritizing the most promising candidates peptides for experimental validation are therefore needed to advance the field of peptide discovery[24,25].

In this study, we conducted a large-scale peptidomics analysis of seven different organs in four different mouse strains. Realizing that the majority of the observed peptide sequences are likely inactive degradation products, we developed a computational algorithm based on machine learning that uses the structure of the peptide clusters observed for each protein to predict the most likely endogenously functional peptides. We show that the algorithm can indeed identify many of the known annotated bioactive peptides directly from our MS data and that the other high-scoring candidate peptides suggested by the algorithm display characteristics similar to those of the known peptides, such as originating from secreted protein precursors and having known cleavage motifs in their flanking regions, even though the algorithm does not use these features as input for its predictions. To further enhance our ability to identify potential bioactive peptides, we combine the output of our prediction framework with in-silico bioactivity prediction based on amino acid composition to identify a number of interesting new candidate peptides. Using in-vitro and in-vivo screening models of diabetes, we confirm potential bioactivity of several of these predicted peptides. For each tissue investigated, we provide a list of the highest scoring peptides that collectively represents a resource ripe for further exploration and functional

characterization. We lastly demonstrate that additional known peptides can be identified by combining our prediction framework with an algorithm that assembles observed degradation fragments into full length peptides.

## Results

### Experimental setup

Our experimental and computational workflow is illustrated in Fig. 1a. We investigated seven metabolically active tissues consisting of liver, muscle, intestine (ileum), brain, pancreas, epididymal fat, and subcutaneous fat from four different mouse strains commonly used in diabetes research; leptin receptor-deficient mice C57bl/KS-*Lepr*<sup>db</sup>/*Lepr*<sup>db</sup> (DB) versus C57bl/KS-*Lepr*<sup>db</sup>/+ (WT), and C57bl/6J mice fed a low-fat diet (LF) or high-fat diet (HF)[26]. To ensure statistical robustness of peptide abundance estimation, we used 12 animal replicates in each group, leading to a total of 336 samples. All tissue samples were heat-stabilized[27,28] before homogenization to minimize post-mortem degradation of the in-vivo peptidome. We elected not to reduce/alkylate the samples as our downstream automated peptide synthesis platform did not cope well with the complexity of intra- or inter-molecular sulfide bridges (Supplementary Fig. 2a). Peptides were subsequently separated from co-purified proteins using a molecular weight cut-off spin-filter[20] which offered scalable and consistent peptide enrichment over proteins without any significant size bias below the cutoff[29,30] (Supplementary Fig. 2b,c). Peptide mixtures were analyzed by online nanoflow liquid chromatography tandem mass spectrometry using high-resolution higher-energy collisional dissociation fragmentation[31]. Rawfiles were searched with a selected number of post-translational modifications using two different search engines which resulted in the high-confidence identification of 157,857 unique peptide sequences across all tissues, conditions, and replicates (Fig. 1b). The output was filtered and combined[32] (Supplementary Data 1), and accuracy, score distribution and sequence coverage were comparable to other peptidomics dataset[20,22] (Supplementary Fig. 3a–j). The data set provide comprehensive coverage of different tissues and conditions under which peptides may be produced and secreted. It is, to our knowledge, the largest single peptidomics study to date in terms of tissue coverage and number of samples/replicates (Fig. 1c).

### Tissue, diet and genetic background manifests at the peptide level

Although many of the observed peptides are found in multiple tissues, each tissue contributes with a unique set of peptides, with ileum being the largest contributor of unique sequences (Fig. 1b). Non-linear dimensionality reduction with UMAP reproduced the tissue and experimental groups (Fig. 2a,b) and clustering showed that tissue type was more strongly manifested in the data than strain background, with the two fat tissues being most closely related (Fig. 2c). Interestingly, different sequence motifs were observed in the flanking regions of the detected peptides from different tissues, in line with the expectation that enzymatic processing varies by tissue (Supplementary Fig. 4). An additional test experiment verified that the tissue-specific peptide signatures do not simply reflect degradation of highly abundant proteins (Supplementary Fig. 5a,b and Supplementary Data 2).

### Sensitivity and degradation in peptidomics data

To assess the sensitivity and coverage of our data, we compared it to a curated list of 294 known and annotated peptides extracted from different peptide databases[33,34] (Supplementary Data 3), and found that 89 (30%) of these appear in our MS data, including their known post-translational modification pattern, with an additional 76 known peptides covered partially by shorter fragments. Our mammalian peptidome analysis of 7 tissues thus covers nearly half of the annotated mouse peptides, as full or partial matches, demonstrating the

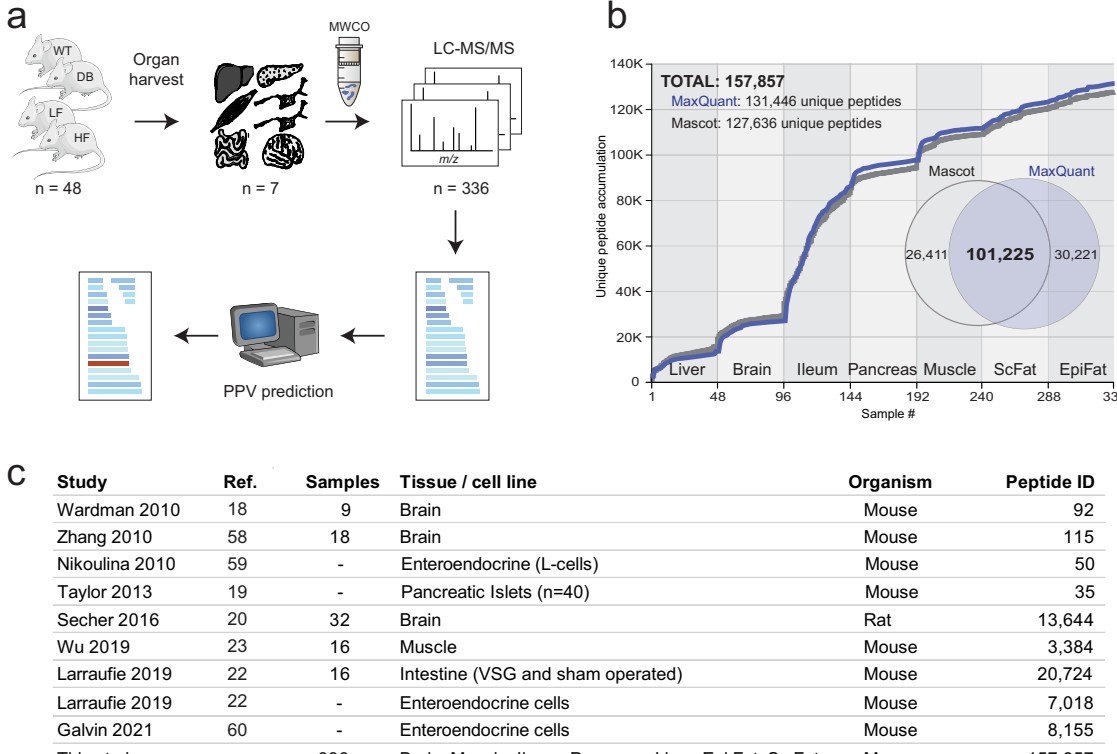

Fig. 1 | **Experimental setup and data overview. a** Age synchronized mice ($n = 12$ each) of wildtype (WT), diabetic (DB), low-fat diet (LF), high-fat diet (HF) background were sacrificed and 7 organs collected. To minimize background interference from protein and peptide degradation, tissue samples were heat stabilized using a Denator T1 heat-stabilizor. Peptides were enriched using a molecular weight cut-off filter (MWCO) and the resulting raw files processed using the Mascot and MaxQuant search engines. The Predicted-Peptide Variant (PPV) algorithm highlight peptides of interest within the data. Mouse and tissue images in this work were produced using Servier Medical Art (https://smart.servier.com/). **b** Accumulated unique peptide identifications from each tissue. The combined dataset consists of 157,857 unique peptides in total. Venn diagram depicts the peptides identified by Mascot (grey) and a combined Maxquant search (blue) when all tissues are searched together. **c** Representative peptidomics studies[18–20,22,23,58–60] conducted in rodents within the last decade.

sensitivity of the technology and the potential of this data set as a resource for discovery of new bioactive peptides. The challenge, however, is that the known peptides are set against a vast background of peptides representing degradation fragments rather than real endogenously produced peptides with biological functions.

Mapping observed peptide sequences onto their cognate protein backbone proved to be a powerful way to illustrate the structure of the data, as exemplified for Secretogranin-1 (CgB); a protein of the granin family known to be processed into bioactive peptides[35] (Fig. 3a). We observe several of the known Secretogranin-1 peptides as well as clusters indicating the existence of other uncharacterized peptides. The peptide clusters observed tend to align perfectly with flanking dibasic motifs[10] (KR, RK, KK, RR, etc.) consistent with Secretogranin-1 being canonically cleaved by prohormone convertases in islets, pancreas, intestine and brain where these enzymes are known to process many other precursors into peptides. Some peptides, like LE-20, are found in multiple tissues, whereas the PE-11 cluster is exclusively observed in brain, in accordance with previous findings[36]. Comparing the pancreatic peptidome (blue) to that of the secreted content from stimulated islets of Langerhans[19] (green) confirms that many of the peptides observed as distinct clusters in whole tissue samples are indeed actively secreted, including LE-20, Bam-1745, and several uncharacterized peptides (Fig. 3a). Our approach even detects intracellularly stored peptides, as demonstrated by the Manserin peptide which is visible in whole pancreatic tissue data but not in our islet secretion data (Supplementary Fig. 6a,b and Supplementary Data 4). Whole tissue peptidomics thus reduces the reliance on knowing and inducing the exact conditions under which secretion

of a given peptide occurs, making it suited for comprehensive discovery-oriented analysis of the peptidome.

Secretogranin-1 (Fig. 3a) at the same time illustrates the challenge of using the data; the observed clusters which contain the known annotated peptides are co-located with observations of many shorter degraded variants making it non-trivial to single out the sequence of the real active variant of the peptide automatically from the data. In some cases, longer unprocessed variants are also seen. On top of this complexity for the known bioactive peptides, routine breakdown of all other proteins is also measured in whole tissue peptidomics.

Using large-scale peptidomics data for discovering potential new bioactive peptides thus present a needle-in-a-haystack challenge of initially separating real functional peptides from the background noise of peptide and protein degradation. Such degradation is not specific to our data but also observed in other recently published peptidomics studies as well (Fig. 3b), even though all of these take preventive measures to minimize inadvertent post-mortem degradation during sample handling. Therefore, new computational methods are needed to separate signals from noise in such data from living cells and tissues.

## Computational identification of bioactive peptides

Since visual inspection, as illustrated for Secretogranin-1 (Fig. 3a), is neither objective, reproducible nor scalable to tens of thousands of proteins and peptides, we developed a computational method to identify the most promising potential new peptides directly from the MS data. We first engineered a series of features from the MS data to represent the positional patterns, such as the relative abundance of peptides starting at a given position relative to the position before it

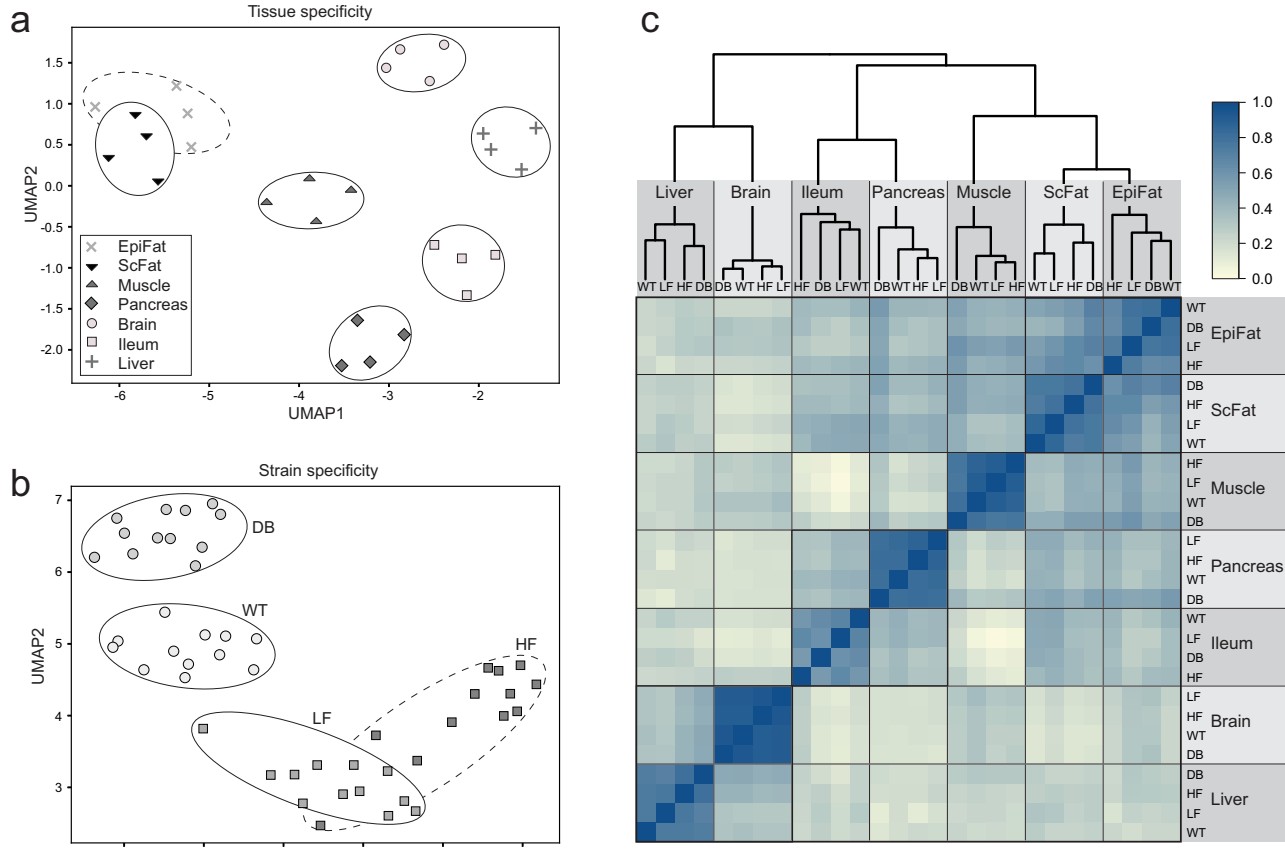

**Fig. 2 | Diet and genetic background manifested at the peptide level. a** Tissue specificity on entire dataset. Non-linear dimension reduction employing UMAP with imputed values on 1% quantile of lowest intensity peptides[61]. Average values from 12 animals for each strain type; wildtype (WT), diabetic (DB), low-fat diet (LF), high-fat diet (HF). **b** Strain specificity using UMAP with only valid values. Each datapoint is one animal. **c** Pearson correlation plot based on all pairwise peptide intensities (abundance) within groups. Median intensity values are used within each $n = 12$ group from the MaxQuant dataset.

(Fig. 4a; intensity start as example), as well as additional features that express amino acid modifications observed by MS (C-terminal amidation and N-terminal acetylation) (see Supplementary Note 1). We then trained various machine learning methods by using the set of known annotated peptides as positive examples (Supplementary Data 3) and the remaining observed peptides as negative examples. Each peptide was represented by its feature vector and nested five-fold cross-validation was used to ensure that the performance metrics reported for each model is based only on predictions on the hold-out test sets unseen by the methods during training and parameter optimization (Fig. 4b, Supplementary Note 1). The more advanced, non-linear models (Random Forest and SVMs) did not confer any advantage over their simpler linear counterparts (Bayesian and simple logistic regression; Fig. 4c, Supplementary Fig. 7a), and no performance benefits were observed with elastic net regularization or SMOTE (Supplementary Fig. 7b,c), despite the highly imbalanced training and test sets (few positive examples and many negative examples). We therefore chose to base our final method, that we term Predicted Peptide Variant (PPV), on simple logistic regression which offers a high degree of explainability and allows easy quantification of the importance and directionality of the input features. We found that the largest contribution to the model predictions comes from the features that encode the likely start and stop of a peptide cluster (Fig. 4b) and that C-terminal amidation adds positively to the predictions, consistent with this modification being required for the bioactivity of many known peptides[37], including Neuromedin-C for which we confirmed the importance of the amidation for the function of the peptide (Supplementary Fig. 8a–c).

We compared the PPV model to a simple null model (AUC 0.732) which uses the abundance of each peptide as its only predictor. The Null model finds only 5 known training peptides among the 300 most abundant peptides, showing that the known peptides cannot be identified simply from their abundance. In comparison, the PPV model (AUC 0.886) identifies 48 known training peptides among its top 300 predictions, corresponding to a 176-fold enrichment over randomly picking peptides from the mass spectrometry data.

The PPV model was used to score and rank all observed peptides in each tissue, resulting in the identification of both known bioactive peptides and uncharacterized candidate peptides. The list of high-scoring peptides for each tissue is available as a supplementary table (Supplementary Data 5) and the source code for the model is openly available from GitHub (https://github.com/jancr/ppv).

It should be noted that although most of the annotated peptides extracted from peptide databases have validated functions, the positive training set (Supplementary Data 3) also contains peptides that are known to exist but for which no bioactivity has yet been reported, such as the C-terminal peptides of NPY and PYY. The PPV method is therefore not a predictor of bioactivity per se but rather a method suited for identifying peptide candidates that manifest with patterns in MS data similar to those displayed by known peptides. We therefore benchmarked the PPV method against a previously published computational method called PeptideRanker, which predicts bioactivity of peptides solely based on their amino acid sequence (Fig. 4c). When applying both methods to our data set, the PPV method identifies 3 times more known peptides (48 vs 14) in the top 300 compared to PeptideRanker and has a better AUC (0.886 vs. 0.769), demonstrating that PPV

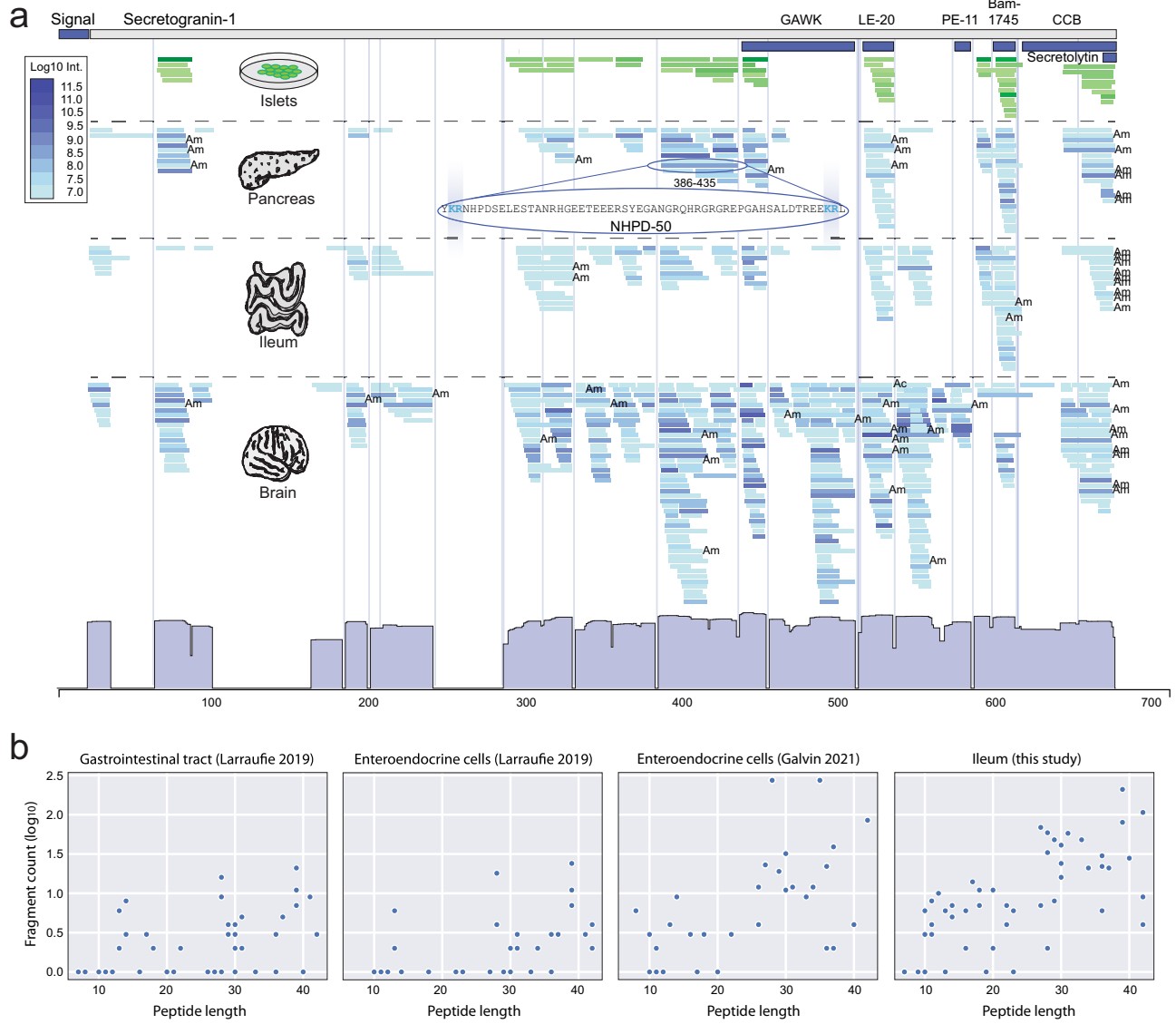

**Fig. 3 | Sensitivity and degradation in peptidomics data. a** The secretogranin-1 protein (Uniprot ID: P16014) shown in grey and previously identified peptides (GAWK, LE-20, PE-11, Bam-1745, CCB) as dark blue boxes. The abundance of the peptides are colour coded green for islets and blue for tissue samples. Abundance histogram shown at the bottom and dibasic motifs depicted by vertical lines. Ac Acetylation, Am Amidation. **b** Degradation characteristics shown as a dot plot for the set of known annotated peptides (Supplementary Data 3) as a function of peptide length (x-axis). The y-axis shows the number of overlapping smaller peptides for the known set ($\log_{10}$ count). Detection of degradation products of known annotated peptides was observed across all studies irrespective of different peptide enrichment methodology applied, illustrates the general needle-in-a-haystack problem of identifying the correct candidate peptide sequence for testing among overlapping fragments. Source data are provided as a Source Data file.

extracts more signal from the structure of the MS-data than PeptideRanker does from the amino acid composition towards recovering the known, annotated – and predominantly bioactive – peptides.

## Model predictions and in-silico validation

To further build confidence in the PPV results, we analysed the flanking regions of the high-scoring predicted peptides and found a clear tissue-specific enrichment of sequence motifs (Fig. 5a). The positive training peptides were removed prior to this analysis to ensure that any signals observed reflect the properties of the new predicted peptides without bias from the peptides that the model was trained on. For brain, pancreas and ileum, the flanking regions show marked enrichment of the di-basic cleavage motifs recognized by prohormone convertases which are known be active in these particular tissues (Supplementary Fig. 9). Different motifs were observed in fat tissues, indicating that other enzymes may be active here. The peptides

predicted by the PPV model also more often originate from secreted proteins (Fig. 5b). This aligns with the fact that most, albeit not all, annotated peptides come from precursors which contain an N-terminal signal peptide. Since none of the input features to the PPV model encode knowledge of the amino acid sequence of the flanking regions around the predicted peptides, or the secretion status of the parent proteins, the enrichment of these properties independently supports the validity of our methodology and predictions.

When applied to tens of thousands of peptides observed in a tissue, the probabilistic PPV model only predicts a small subset of peptides with a high probability of being real peptides (Fig. 5c). The number of very high-scoring (>0.05; red) and high-scoring (>0.01; yellow) peptides were 93/610 for brain, 39/481 for intestine (Ileum), 64/455 for pancreas, 20/229 for liver, 18/306 for quadriceps muscle, 34/346 for epididymal fat, and 42/462 for subcutaneous fat, respectively (Supplementary Data 5). Since the PPV method is not a perfect

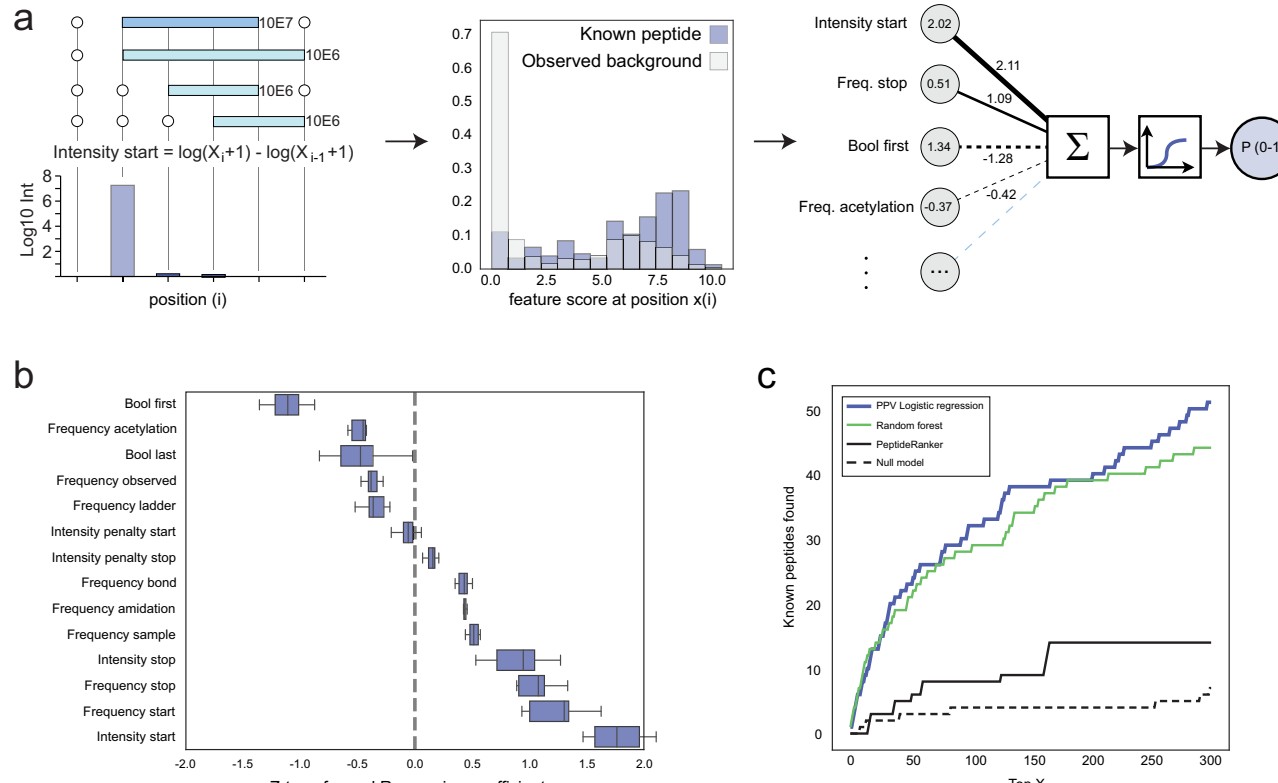

**Fig. 4 | Computational identification of potential bioactive peptides.**
**a** Illustration of feature encoding and distribution (using the feature "Intensity start" as example) and PPV model design (Supplementary Note 1). In the feature distribution plot, the group of known annotated peptides is shown as blue bars, all other observed peptides as white bars and the overlap between the two in light blue. **b** Box plot showing the model regression coefficients (weights) in the final model. Each box-plot shows the coefficients across the 20 models from 5-fold nested cross-validation. The box plot centre line is the median, the bounds are the lower (25th percentile) and upper (75th percentile) quartile values. The lower whisker extends to the lowest observed value greater than the lower quartile minus 5 times the interquartile range (IQR) of the data, the upper whisker to the highest observed value lower than the upper quartile plus 5 times the IQR. **c** Number of known annotated peptides found by each type of model as a function of rank based scoring, using nested cross-validation. PPV model based on Logistic Regression (solid blue), Random forest (green), PeptideRanker (solid black), Null model (dotted black); model uses only a single input feature, the $\log_{10}$ (total peptide abundance). See Supplementary Fig. 7 and Supplementary Note 1 for description of the different models, features and model development. Source data are provided as a Source Data file.

predictor and since it does not predict bioactivity per se, we should expect some of the predicted peptides to be false positives or real peptides with no direct biological activity (e.g. the C-terminal peptides of NPY and PYY). We therefore advice the reader to take into consideration other data as well when evaluating these lists which are purely based on signals extracted from the MS peptidomics data.

Table 1 illustrates the predictions for the 25 highest scoring peptides found in brain tissue demonstrating the PPV method's ability to identify known bioactive neuropeptides among tens of thousands of other peptides observed. Many of the predicted peptides identified share the flanking sequence motifs characteristic of neuropeptides (recognized by pro-hormone convertases).

The PPV predictions are illustrated visually for the neurosecretory protein VGF (Fig. 5d) using red and yellow colour to highlight the highest scoring peptides. The data and predictions confirm the existence of previously reported peptides but also shows that VGF processing is considerably more complex than hitherto described. An example is the brain-specific and highly conserved, 46-amino acid peptide (position 375-420) which we decided to call "GGGE-46" (Fig. 5d). This peptide is among the high-scoring in brain (PPV score 0.017) and its sequence is perfectly flanked by di-basic cleavage motifs (RR and KR). Interestingly, a previously reported shorter form (375–407)[38] scores well below our threshold, illustrating the discriminatory power of the PPV algorithm. The model also correctly identifies the right variant of NERP-1 (rank 10 in brain with score 0.273)

which is involved in body fluid homeostasis[39] (in both brain and pancreas independently) and confirms the existence of the uncharacterized "24-63" peptide (rank 49, score 0.084), while scoring other variants within those clusters low. Both peptides were only annotated recently and therefore not included as positive examples during training (in fact, they were wrongly labelled as negatives). Their identification among the top 50 highest scoring peptides in brain (out of 20,597 observed) thus serve as independent "in-silico validation" of the PPV model's ability to generalize beyond its training data (no overfitting). The correct identification of NERP-1 furthermore demonstrates that the model does not simply pick the longest variant in a cluster, since several low scoring variants were found to extend beyond the NERP-1 sequence. Other high-scoring in-silico validation examples that were not part of the positive training set include Serpinin-RR, 'joining peptide' from POMC, EH24 from pro-Thyrotropin-releasing-hormone and IP2 from pre-pro-glucagon (Supplementary Data 5).

Another way to visualize and explore the data is to plot the results of the two orthogonal machine learning approaches (PPV based on MS data and PeptideRanker based on sequence composition) against each other for the highest scoring 300 peptides (Fig. 5e) using different coloring for the known peptides and the uncharacterized and potentially bioactive peptides. This nicely illustrates that many of the peptides identified via their expression pattern in our large MS data set (by the PPV method) have an amino acid composition similar to that of the

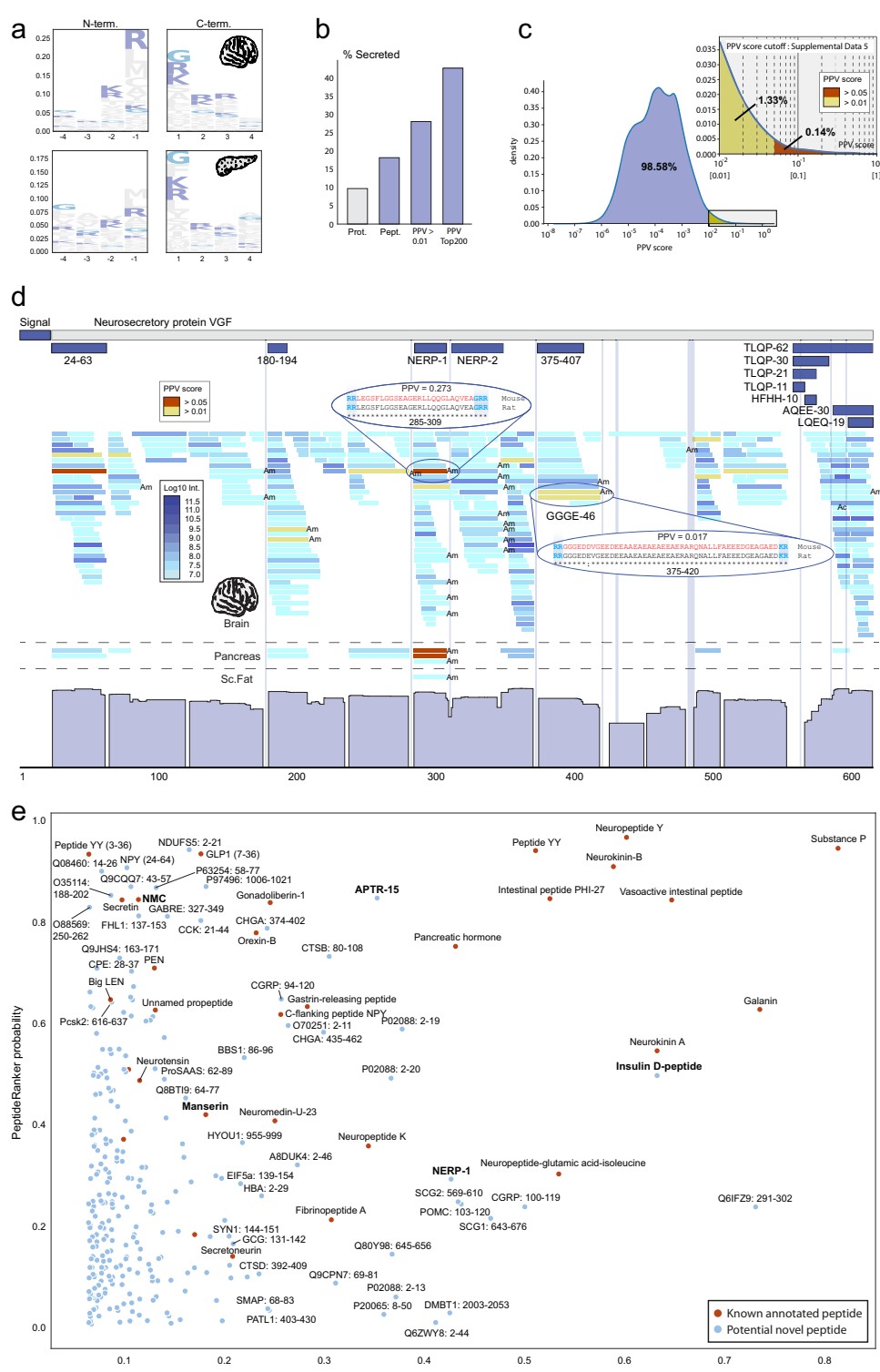

**Fig. 5 | PPV model validation and predictions. a** Logo-plot of N- and C-terminal flanking regions surrounding high scoring PPV predicted peptides in brain and pancreas. Lysine (K) and Argenine (R) are marked in dark blue, and Glycine (G) in light blue. Calculated by Kullback-Leibler divergence between PPV predictions with a score above 0.01 and the 80% lowest scoring background PPV predictions. All known annotated peptides were removed. **b** Secretion status determined by membership on protein precursor level based on signal peptide annotation and signalP predictions. Mouse proteome (Prot.), the global peptidome (Pept.) data in this study, PPV predictions with scores above 0.01 and Top200 PPV predictions across all tissues. All known peptides were removed. **c** PPV score density distribution in the data. X-axis is the PPV-score ($Log_{10}$), and the

Y-axis is number of peptides. **d** Neurosecretory protein VGF (Uniprot ID: Q0VGU4) and tissue specific peptide patterns. Peptides are colour coded according to $Log_{10}$ abundance (blue) and high-scoring predicted PPV peptides highlighted (yellow: >0.01, red: >0.05). Previously reported VGF peptides are mapped onto the cognate backbone at the top based on sequence similarity to annotated rat peptides. Ac Acetylation, Am Amidation. **e** PeptideRanker predictions (amino acid sequence composition) for the top300 PPV predictions (mass spectrometry signal) plottet against each other. Known annotated peptides used in the PPV training marked in red, unknown (negative set) marked in blue. Source data are provided as a Source Data file.

## Table 1 | PPV predictions in brain

| PPV | ID | Gene | Peptide | N- | Peptide Sequence | Am. | C- | Sec. | Tissue(s) |
|---|---|---|---|---|---|---|---|---|---|
| 0.670 | P41539:58-68 | Tac1 | Substance P [T] | RIAR | RPKPQQFFGLM | Yes | GKRD | Yes | B,I,P |
| 0.588 | P55099:82-91 | Tac3 | Neurokinin-B [T] | PQKR | DMHDFFVGLM | Yes | GKRN | Yes | B |
| 0.533 | P56942:131-143 | Pmch | Neuropeptide-Glu-Ile [T] | QEKR | EIGDEENSAKFPI | Yes | GRRD | Yes | B |
| 0.532 | P41539:98-107 | Tac1 | Neurokinin A [T] | SHKR | HKTDSFVGLM | Yes | GKRA | Yes | B,I,P |
| 0.455 | P47212:33-61 | Gal | Galanin [T] | KEKR | GWTLNSAGYLLGPHAIDNHRSFSDKHGLT | Yes | GKRE | Yes | I,B,Ep,P |
| 0.436 | P01193:103-120 | Pomc | Joining peptide (mis-annotated) | AQRR | AEEEAVWGDGSPEPSPRE | Yes | GKRS | Yes | B |
| 0.383 | P32648:81-107 | Vip | Intestinal peptide PHI-27 [T] | RNAR | HADGVFTSDYSRLLGQISAKKYLESLI | Yes | GKRI | Yes | B,I,P |
| 0.317 | P32648:125-152 | Vip | Vasoactive intestinal peptide [T] | PIKR | HSDAVFTDNYTRLRKQMAVKKYLNSILN | Yes | GKRS | Yes | P,B,I |
| 0.295 | Q03517:569-610 | Scg2 | | VSKR | IPVGSLKNEDTPNRQYLDEDMLLKVLEYLNQEQAEQGREHLA | No | KRAM | Yes | B,P,I,Sc,Ep |
| 0.273 | Q0VGU4:285-309 | Vgf | NERP-1 (delayed annotation) | KVRR | LEGSFLGGSEAGERLLQQGLAQVEA | Yes | GRRQ | Yes | B,P,Sc |
| 0.264 | P41539:72-107 | Tac1 | Neuropeptide K [T] | MGKR | DADSSVEKQVALLKALYGHGQISHKRHKTDSFVGLM | Yes | GKRA | Yes | B,I |
| 0.244 | P13562:22-31 | Gnrh1 | Gonadoliberin-1 [T] | GCSS | QHWSYGLRPG | Yes | GKRN | Yes | B |
| 0.243 | Q3TC46:403-430 | Patl1 | | DPYA | NLMLQREKDWVSKIQMMQLQSTDPYLDD | Yes | FYYQ | No | B |
| 0.230 | O55241:69-96 | Hcrt | Orexin-B [T] | LGKR | RPGPPGLQGRLQRLLQANGNHAAGILTM | Yes | GRRA | No | B |
| 0.212 | P57774:68-97 | Npy | C-flanking peptide of NPY [T] | YGKR | SSPETLISDLLMKESTENAPRTRLEDPSMW | No | ---- | Yes | B,I,P,Ep,Sc,Q |
| 0.203 | O88935:144-151 | Syn1 | | DIKV | EQAEFSDL | Yes | NLVA | No | B |
| 0.199 | B2RXS4:1131-1154 | Plxnb2 | | MTLE | EAEAFVGAERCIMKTLTETDLYCE | Yes | PPEV | No | B |
| 0.198 | Q99JA0:100-119 | Calca | Calcitonin GR peptide (short form) | GLLS | RSGGVVKDNFVPTNVGSEAF | Yes | GRRR | Yes | S,B,Ep,Q,I |
| 0.190 | Q62361:25-50 | Trh | N-terminal fragment (unannotated) | KSCA | LLEAAQEGAVTPDLPGLEKVQVRPE | No | RRFL | Yes | B |
| 0.160 | A8DUK4:2-46 | Hbbt1 | | _M_ | VHLTDAEKAAVSGLWGKVNADEVGGEALGRLLVVYPWTQRYFDSF | Yes | GDLS | No | B,Q,Sc |
| 0.159 | O70251:2-11 | Eef1b | | _M_ | GFGDLKTPAG | Yes | LQVL | No | B,L,I,P |
| 0.157 | P57774:29-64 | Npy | Neuropeptide Y [T] | LAEG | YPSKPDNPGEDAPAEDMARYYSALRHYINLITRQRY | Yes | GKRS | Yes | E,B,Sc,I,P |
| 0.153 | P09240:21-44 | Cck | Propeptide (short form) | GALA | QPVVPAEATDPVEQRAQEAPRRQL | No | RAVL | Yes | B,I,P |
| 0.141 | A2AWN3:327-349 | Gabre | | EPKP | EPEPQPEPEPEPEPEPQPEPE | Yes | PKPE | No | B |
| 0.141 | Q9D3D9:23-46 | Atp5d | | ARTY | AEAAAAPAPAAGPGQMSFTFASPT | Yes | QVFF | Yes | B |

Top 25 highest scoring PPV peptides in brain. Uniprot protein identifier is separated from peptide position with a colon. The peptide column contains annotation status or a suggested name (SN). If followed by a [T] the peptide was included as positive training example. Other named peptide was not used for training but later found to match a previously described peptide (either due to delayed annotation or mis-annotation). Dibasic cleavage motifs are indicated in the N- and C-terminal flanking regions, with potential Glycines in position +1. C-terminal Amidation (Am.: Yes if >98% and No if <2%). Secretion status is indicated with Yes/No and tissue abbreviations are as follows; Brain (B), Ileum (I), Pancreas (P), Liver (L), EpiFat (Ep), ScFat (Sc), and Quadriceps muscle (Q).

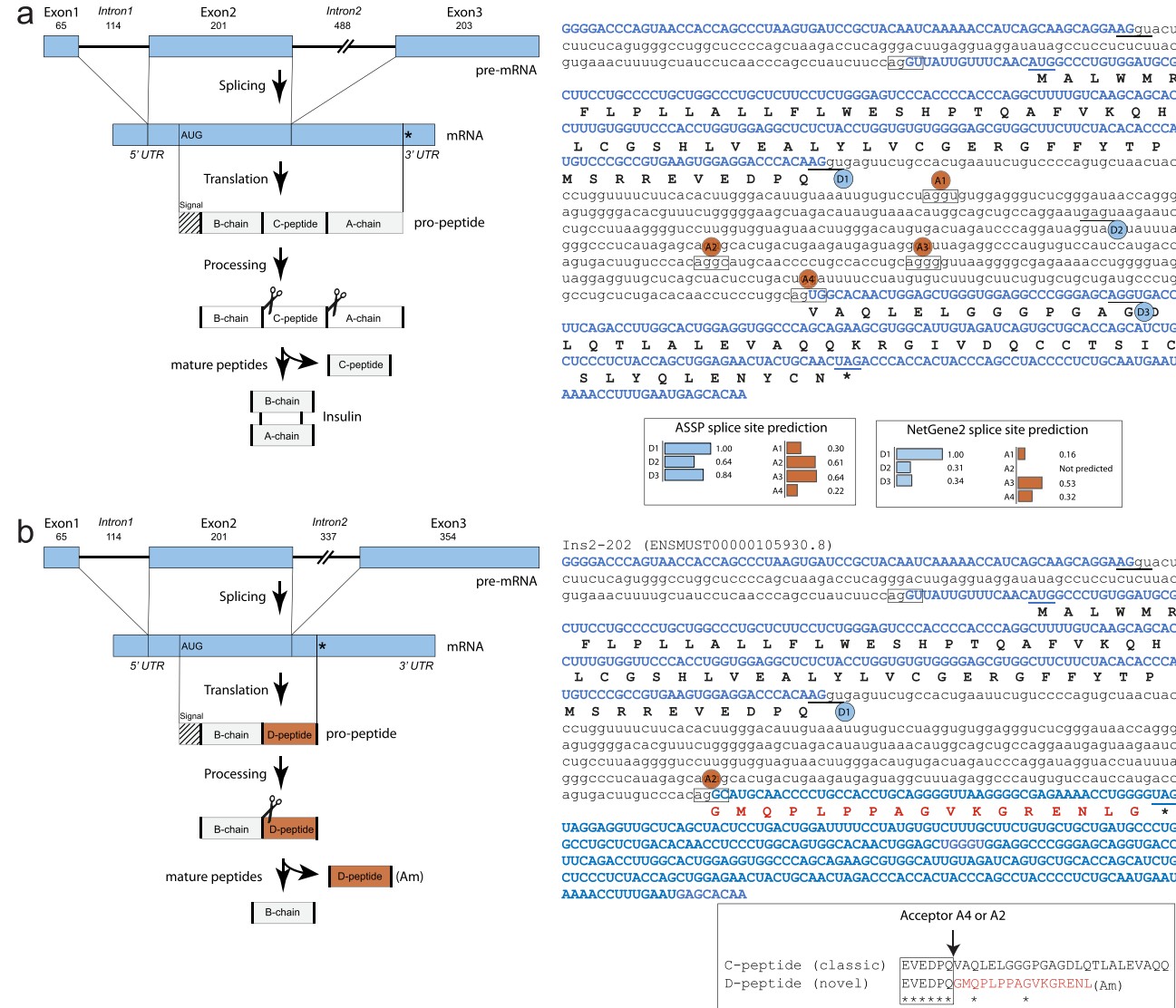

**Fig. 6 | A unique high-scoring peptide formed through alternative splicing in insulin. a** Insulin-2 (Uniprot: P01326; NCBI sequence: NC_000073.7: c142233463-142232393) from *Mus musculus* strain C57BL/6J. Splicing and pro-peptide processing will generate the mature insulin's A- and B-chain and the connecting-peptide (C-peptide). The splicing in intron2 takes place between donor site 1 (D1) and acceptor site 4 (A4) as shown on the right. Introns are in small letters, exons is in capital letters. The ASSP and NetGene2 splice site predictions show the relative strength of donor sites (underlined; blue) and acceptor sites (boxed; red) normalized to D1.

**b** The Ins2-202 transcript (Ensembl ID: ENSMUST00000105930.8) match the Uniprot entry D3Z596, and is a 79 amino acid variant of insulin. Alternative splicing between the D1 donor site and the cryptic A2 acceptor site introduces a new reading frame with a premature stop-codon forming a pro-peptide consisting of an intact B-chain and a uncharacterized peptide (D-peptide) after prohormone convertase cleavage. The PAM enzyme converts the terminal Glycine in 2 sequential steps to a C-terminal amidation. Bottom right contains an alignment between the classic C-peptide and the D-peptide.

known bioactive peptides (high PeptideRanker scores), which increases the likelihood that these are in fact bioactive.

## Examples of uncharacterized peptides identified

Much to our surprise, the highest-scoring peptide in pancreas tissue (Supplementary Data 5; Pancreas) is a hitherto undescribed peptide found exclusively in pancreatic tissue where it is observed in all 48 mice (4 conditions x 12 replicates). The 22-amino acid peptide maps to an unreviewed protein precursor (Uniprot: D3Z596) originating from the insulin-2 gene. Its N-terminal sequence is identical to that of the connecting peptide (C-peptide) of insulin (including the known cleavage motif) but the rest of the sequence is distinctly different, and the peptide is amidated at its C-terminal. The sequence matches neither the normal insulin-2 precursor (P01326) nor that of insulin-1 (P01325), and we therefore considered if alternative splicing could explain our findings.

A splice site prediction analysis of the second intron of Insulin-2 suggests several possible acceptor sites (Fig. 6a), however only through splicing to a cryptic A2 acceptor site would the correct reading frame be established (Fig. 6b). This particular reading frame includes a C-terminal Glycine right before the stop-codon which is likely converted by the PAM enzyme to the C-terminal amidation we observe in our data (Supplementary Fig. 10a–c). This suggests a possible secretory role for the alternatively spliced peptide, that we termed Disjoining-peptide (D-peptide). Curiously, the D-peptide is scored 36 times higher than the classical C-peptide by the PPV algorithm, indicating that the D-peptide, not the C-peptide, is the one which stands out most clearly in the MS-data. The D-peptide is also scored relatively high by PeptideRanker (0.495) in support of a potential bioactive role for the peptide.

Some of the peptides are found independently in multiple tissues and we therefore combined the prediction probabilities from

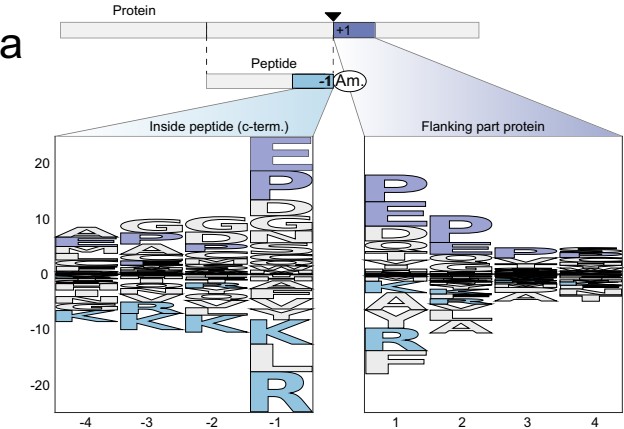

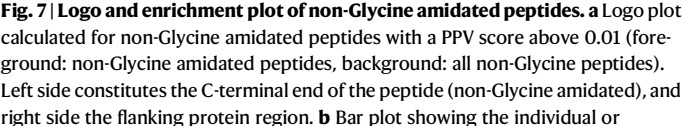

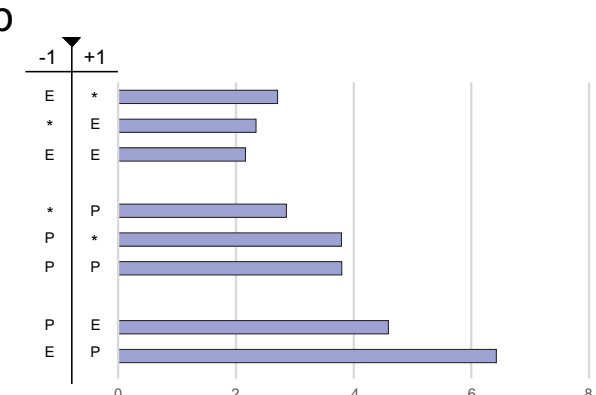

**Fig. 7 | Logo and enrichment plot of non-Glycine amidated peptides. a** Logo plot calculated for non-Glycine amidated peptides with a PPV score above 0.01 (foreground: non-Glycine amidated peptides, background: all non-Glycine peptides). Left side constitutes the C-terminal end of the peptide (non-Glycine amidated), and right side the flanking protein region. **b** Bar plot showing the individual or combinatorial enrichment of amino acids in the −1 and +1 position around the non-Glycine amidated peptides for PPV predictions above 0.01. The length of the bar is the enrichment of the motif compared to the background. Source data are provided as a Source Data file.

individual tissues into a consolidated score for each peptide across all tissues, as an alternative way of viewing the prediction output (Supplementary Data 5; PPV combined). This approach highlighted an N-terminal peptide derived from the glycolytic enzyme Tpi1 that the PPV method identifies consistently across brain, pancreas, muscle and fat tissues. The peptide, that we named APTR-15, contains the substrate binding residues of Tpi1 and is structurally followed by an exposed loop containing a prohormone convertase GRK motif consistent with its observed C-terminal amidation (Supplementary Fig. 11a–c). It should be noted that the enzyme is intracellular and would not under normal circumstances be co-localized with the prohormone convertases in the secretory pathway. Our data nonetheless strongly indicate that this particular peptide is formed in-vivo across multiple tissues. A potential bioactive role for this peptide is supported by PeptideRanker which scores it higher (0.844) than well described peptides like pancreatic hormone, Orexin-B, Gastrin-releasing peptide and VIP (Fig. 5e, Supplementary Data 5).

**Non-Glycine mediated amidation**
Some of the high scoring peptides are found (in our MS data) to be amidated at the C-terminus without the Glycine (in position +1) that would normally be recognized and converted by the PAM enzyme (Table 1, Supplementary Data 5). Such PAM-independent peptide amidation has been reported in several other studies[20,21] and detailed inspection of the MS/MS ion spectra of selected examples from our study indeed support C-terminal amidation (Supplementary Fig. 12a–h). We cannot completely exclude the possibility of side-chain Glutamic acid to Glutamine conversion among a smaller subset of Glutamic acid containing peptides where the fragment ion series does not unambiguously confirm the amidation to be positioned in the C-terminal. A closer look at the sequence context around these non-Glycine amidations (Fig. 7a) reveal a decrease in dibasic amino acids, indicating that they are likely not products of prohormone convertases in the canonical secretory pathway. These peptides are instead enriched in a E/P motif around the site of amidation (Fig. 7a,b). These findings apply across the global data set as well as among those peptides selected by our algorithm. Although the enzymatic or non-enzymatic amidation mechanism remains to be elucidated, the data indicates that the majority of these peptides cannot be explained as random artifacts of the experimental procedures or the algorithm.

**Experimental validation of predicted peptides in diabetes-related models**
The computational framework developed and presented in this study can be used to shortlist the most promising peptides of which many should statistically be expected to be bioactive (Supplementary Data 5,6). Validating such bioactivity and uncovering the mode of action of a candidate bioactive peptide is, however, inherently difficult, as it requires some initial functional hints to guide the choice of assays or models for confirmation. Such hints could be obtained from specialized sequence-based prediction tools like MultiPep[40] although they still predict fairly broad functional categories (e.g. "neuropeptide") that may have limited utility for guiding assay selection.

To authenticate the bioactivity of some of our predicted peptides, we relied on a set of diabetes-related assays and models already established in house and validated using known metabolic peptide hormones (Supplementary Fig. 13a–d) and selected non-functional peptides (Supplementary Fig. 14a–d). We therefore sought to increase the chance of a functional read-out in these particular assays by selecting candidates that originate from pre-cursor proteins/genes with functional links to diabetes, obesity and metabolism in public databases such as Gene Ontology (geneontology.org)[41], DISEASES (diseases.jensenlab.org)[42] and PHAROS (pharos.nih.gov)[43]. This approach obviously has limitations in cases where such information is scarce or where the function of the peptide is far removed from that of the parent protein/gene-level to which such data is linked (Supplementary Fig. 1).

Selected peptides were chemically synthesized or expressed in *E. coli* and tested for activity in-vitro, as well as injected subcutaneously in *db/db* mice to quantify acute changes in relative blood glucose (BG) in-vivo. One of the predicted peptides that displayed potential bioactivity was a previously undescribed peptide from Secretogranin-1 that we termed NHPD-50 (position 386-435). It has a high PPV score (0.028) and is flanked by dibasic KR-motifs (Fig. 3a, Supplementary Data 5). Secretogranin-1 (CgB) is a component of the neuroendocrine secretory granules, and peptides from the granin family are known to regulate catecholamine, islet amyloid polypeptide and insulin release[44]. The NHPD-50 peptide reduced BG significantly four hours after single dose administration in *db/db* mice (Fig. 8a) and a significant increase of insulin in plasma was observed six hours after administration (Fig. 8b). The finding was replicated in a follow-up experiment using twice daily dosing (Fig. 8c) which showed a sustained BG lowering effect of this peptide over 8 h. In contrast, injection of low scoring peptides did not elicit any response in these in-vivo models (Supplementary Fig 14a–d).

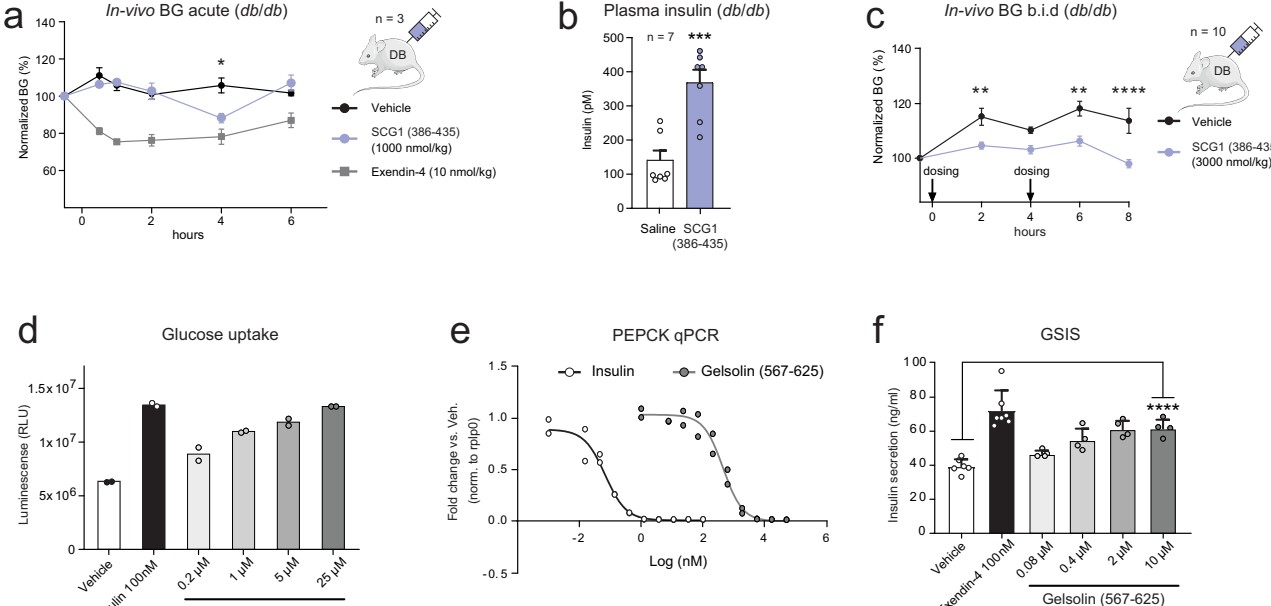

**Fig. 8 | In-vitro and in-vivo assays reveals potential bioactive peptides. a** BG mean values ± SEM in n = 3 *db/db* mice after SCG1 peptide (NHPD-50, position 386–435) intraperitoneal injection at 1000 nmol/kg. Exendin-4 at 10 nmol/kg. Baseline normalized to 100%. Two-way ANOVA with post-hoc Dunnett's test, *P* = 0.022 (4 h). **b** Plasma insulin (pmol/L) mean values ± SEM in n = 7 *db/db* mice 6 h after SCG1 peptide (NHPD-50) dosing at 2658 nmol/kg. Unpaired two-sided t-test, *P* = 0.0003. **c** BG mean values ± SEM in n = 10 *db/db* mice by twice daily SCG1 peptide (386–435) dosing at 3000 nmol/kg. 8 h of day 1 are shown. Two-way ANOVA, Sidak's Multiple comparisons test, *P* = 0.0055 (2 h), *P* = 0.0013 (6 h), *P* < 0.0001 (8 h). **d** Glucose uptake in 3T3-MBX cells at day 19. *N* = 20,000 cells/well in two independent biological experiments. Cells were treated overnight with diluted Gelsolin peptide (567–625) or with 100 nM Insulin. **e** Pck1 qPCR in primary rat hepatocytes. *N* = 30,000 cells/well in two independent biological replica experiments. Cells treated for 4 h with serial diluted Gelsolin peptide (567–625) or serial diluted Insulin. Pck1 expression was normalized to Rplp0. **f** Glucose-stimulated insulin secretion (GSIS) calculated as ng/mL ± SD in INS-1E luciferase cells (50,000 cells/well). The cells were incubated with Gelsolin peptide (567–625) or Exendin-4 at 100 nM for 30 min. Vehicle and Exendin-4 controls are n = 6 independent measurements, whereas Gelsolin treated cells are n = 4 independent measurements. Unpaired two-sided t-test, *P* < 0.0001. Source data are provided as a Source Data file.

We did not observe a glucose stimulated insulin secretion (GSIS) response in INS1E cells or in mouse primary islets suggesting that the effect on blood glucose and the peptide-mediated plasma insulin increase acts through an indirect mechanism. To explore if an engineered NHPD-50 peptide variant with longer half-life could induce sustained BG reduction, we generated an N- and C-terminal albumin protracted version and expressed the construct in *db/db* mice using hydrodynamic gene delivery (Supplementary Fig. 15a). The protracted peptide did not confer sustained BG lowering effect in this experiment and offered only limited effects on plasma triglycerides and free fatty acids (Supplementary Fig. 15b,c). We did, however, observe a significant reduction of 3-hydroxybutyric acid in plasma suggesting a potential role in fatty acid metabolism (Supplementary Fig. 15d). In summary, our experiments support that the predicted NHPD-50 peptide displays bioactivity, although the data is not sufficient to decipher the exact biological mechanism of the new peptide.

Despite attempts to discover the potential biological function of the alternatively spliced insulin D-peptide, we did not observe any acute changes in BG, suggesting that the D-peptide probably maintains other functions unrelated to insulin or the classical C-peptide[45].

### Fragment assembly identifies known and uncharacterized peptides

For some of the known bioactive peptides without a perfect full-length match, we observed shorter peptides that collectively covered the sequence. We therefore devised an algorithm which assembles overlapping fragments into in-silico predicted full-length peptides and found an additional 17 matches to known peptides including Amylin, Glicentin, Gastrin-releasing peptide, Somatostatin-28, PACAP-related-

peptide and the CART peptide, demonstrating the potential of fragment assembly for identifying real bioactive peptides.

We therefore combined the assembly algorithm with the PPV model into a method termed PPV-assembly (Supplementary Note 1) with the aim of discovering additional peptides from our data (Supplementary Fig. 16a, b). The assembly creates on average 5 extra synthetic peptides for every observed peptide (expanding the data set from ca. 150.000 sequences to 1.2 million in total), many of which are likely not real (Supplementary Data 6) and we therefore found the PPV-assembly approach most useful in cases where other sources of information points to a particular protein of interest.

As an example, the model combined fragments in-silico to predict a new peptide in Gelsolin (Supplementary Fig. 17). Gelsolin is a calcium-regulated protein secreted into the blood facilitating pancreatic β-cell proliferation and insulin secretion through actin remodelling and syntaxin-4 interaction[46]. Its N-terminal domain (28–161) has been suggested to normalize BG in diabetic mice[47]. When administering this PPV-assembled peptide (position 567–625) to 3T3L1-MBX cells, we observed a dose-dependent increase in glucose uptake (Fig. 8d) and a dose-dependent inhibition of PEPCK expression in primary hepatocytes, indicative of gluconeogenesis suppression (Fig. 8e). Furthermore, GSIS (insulin secretion) in INS1E cells was increased at the highest concentrations (Fig. 8f). The in-vitro data thus suggests that the predicted in-silico assembled Gelsolin peptide may have a biological function. However, neither the new peptide (567–625) nor the N-terminal domain (28–161) affected BG in our in-vivo model and the possible in-vivo mechanism of action for the peptide thus remains unclear.

## Discussion

The complex task of discovering new bioactive peptides is a two-step process, as one needs first to identify likely bioactive candidates and secondly employ the correct assay to unequivocally authenticate bioactivity.

In this study, we provide a resource to support the first step by sharing lists of candidate peptides (observed and assembled) predicted to be real and potentially bioactive by our machine learning framework (the PPV method) which we apply to a large whole-tissue peptidomics study of seven tissues that—to our knowledge—is by far the largest study conducted to date in terms of tissues, replicates and samples. The benefits of this approach are described above but here we also wish to highlight the limitations and biases of the resource we share with the broader scientific community.

Firstly, many important tissues and biological conditions are not covered in our study, meaning that certain peptides may be missed entirely. Secondly, we initially tested the use of reduction and alkylation but found that this only provided marginal improvements in the identification of cysteine-containing peptides at the expense of poorer identification rates of other peptides. Taking into account the need to disentangle intramolecular cysteine-bridges and the difficulty in chemically synthesizing such peptides, we chose to not reduce and alkylate. Some known cysteine-containing peptides, like insulin, are thus absent from our analysis. The same applies to peptides with certain post-translational modifications[48], like the active form of CCK8 which is sulfated on Tyrosine. To avoid a combinatorial explosion in the search space, the MS analysis was performed with only Oxidation (M), acetylation (N-term), amidation (C-term) and pyro-glu (Q/E) as variable modifications. The PPV method does identify the correct unmodified variant of CCK8 illustrating both the strength and limitation of the approach, as well as the difficulty in authenticating bioactivity which depends on testing the correct bioactive form in the right assay.

The machine learning based algorithms (PPV, PeptideRanker, etc.) are also not perfect, and should be expected to produce both false negatives (bioactive peptides missed) and false positives (non-functional peptide variants and degradation products predicted to be real and/or bioactive). The chance of the latter is arguably lower among peptides originating from secreted proteins and/or those precursors already known to form other bioactive peptides. We therefore provide the readers with the option to filter on these properties in the lists of candidate peptides (Supplementary Data 5,6). Vice versa, more false positives should be expected among predicted peptides originating from cytosolic proteins and we also note that the PPV model may have a slight bias towards peptides that start right after an initiator Methionine. Nonetheless, we found it relevant to share all predictions, since some bioactive peptides are known to be produced from intracellular proteins (Supplementary Fig. 1).

Another limitation of our approach is that both machine-learning frameworks used in this study (PPV and PeptideRanker) are trained to identify bioactive peptides as a general class, and hence cannot be used to pinpoint the exact biological- or molecular function of a peptide to guide assay selection. We did not in this work explore prediction of more fine grained functional categories owing to the relatively small set of positive examples (bioactive peptides) available to train the machine learning framework on.

In this work, we used other contextual information to select candidate peptides to test in a set of assays available to us. A larger systematic study would be required to assess the validity of this approach as a generalized concept for matching candidate peptides and functional assays.

The experimental validation of bioactivity also comes with its own set of limitations. Even in cases where the right bioactive form of a peptide is identified and tested in a suitable model, the functional readout may still be absent due to formulation issues, short peptide half-life, or the absence of other complementary factors required for full activity.

The central idea behind our PPV method is to turn the degradation "noise" that characterizes MS-based peptidomics studies into an asset rather than a challenge by learning from the global and local structure of the data. The concept could potentially be improved with new or improved feature encoding, unsupervised representation learning, and more refined approaches for assembling peptides in-silico.

In conclusion, our peptide data and predictions provide a comprehensive resource amenable for functional exploration beyond diabetes and suggest that more peptides than previously recognized could be functional entities with potential regulatory roles.

## Methods

### Animal experiments

Animal studies were carried out in accordance with the Danish Act on Experiments on Animals - Appendix A of ETS 123 and EU Directive 2010/63. Animal protocols was approved by the Institutional Animal Care and Use Committee of Novo Nordisk Research Center China and Ethical Review Council in Novo Nordisk A/S (Permission no. 2015-15-0201-00616). Animals were purchased either from Taconic (lean C57bl/6J mice), Jackson Laboratory (diet induced obese C57bl/6J mice fed 60% high fat diet (HF) and C57bl/6J mice fed 10% low fat diet (LF)) or Charles River (C57bl/KS *db/db* and C57bl/KS *db/+*). Mice were all male and 12 weeks of age upon arrival and housed under standard conditions including a 12 h–12 h light–dark cycle, ~21 °C, and water and food *ad libitum*. All mice were anaesthetized using isoflurane/$O_2$/$NO_2$ and following tissue extraction euthanized by decapitation at 15 weeks (C57bl/KS *db/db* "DB" and C57bl/KS *db/+* "WT") or at age 26 weeks ("HF" and "LF" mice). Collected organs (subcutaneous fat, epidydimal fat, pancreas, brain, liver, quadriceps muscle and gut ilium) were quickly rinsed in ice cold PBS and freeze clamped with Nitrogen-cooled iron. For the perfusion experiment anesthetized animals were transcardially perfused with an isotonic saline solution containing protease inhibitors[20]. Removed organs were flash frozen in 2-methyl-butane and stored at −80 °C. Male diabetic BKS.Cg-*Dock7^m +/+ Lepr^db*/J (stock no: 000642) strain introduced from the Jackson Laboratory USA was used for blood glucose measurements. Upon arrival, mice were housed as 5 mice per cage under a condition of 12 h light/dark cycle with controlled temperature (26 ± 2 °C), controlled humidity (55 ± 10%) and free access to water and chow diet (Altromin 1320 diet).

### Islet purification and stimulation

500 islets pr. replica experiment were isolated from mouse pancreas using an established protocol[49]. The islets were conditioned in serum-free medium (Gibco) containing 2 mM L-glutamine with Pen/Strep. The islets were stimulated with 500 μM 3-isobutyl 1-methylxanthine (IBMX) and 10 μM Forskolin for 1 h at 37 °C before being collected at 290 g for 2 min. The supernatant was analyzed for the secreted peptide content.

### Peptide extraction

Removed organs were flash frozen in 2-methyl-butane and stored at −80 °C. For analysis tissue samples were taken directly from a frozen state and heated to 95 °C in an air-evacuated cartridge in a Denator T1 heat stabilizor[27]. Extraction of peptides was done essentially as (ref. 20) with the following modifications. We used 5 μL mg$^{-1}$ tissue weight of 0.5% acetic acid or 6 M urea as determined to be most effective for each individual tissue. Microcon YM-10 cut-off filters (Millipore) were preconditioned with 500 μL 2% MeCN/3% MeOH and 500 μL 5 M urea in 0.5% acetic acid. The filtrate was loaded onto in-house packed reverse-phase C8 STAGE tips with two Empore C8 discs preconditioned with

40 μL MeOH, 40 μL 80% MeCN/0.5% acetic acid and twice with 50 μL 0.5% acetic acid/0.1% trifluoroacetic acid (TFA). Stage tips were washed twice with 200 μL 0.5% acetic acid/0.1% TFA.

## Shotgun proteomics

For brain proteomics experiments $n = 3$ of each strain were used. Proteins were extracted with 6 μL/mg of 6 M Gnd-HCl, 5 mM TCEP, 10 mM CAA, 10 mM Tris-Cl pH = 8.0, 1 mM EDTA lysis buffer after denaturing heat stabilization of the brain sample. One mg protein from each sample was reduced/alkylated and subsequently digested with 5 μg Lys-C for 2 h and 5 μg Trypsin in 1% SDC, 100 mM Tris-Cl pH 8.0 overnight at 37 °C using a FASP approach with a 10 kDa MWCO spin filter. The tryptic peptides were subsequently cleaned up on a C18 SepPak column using standard methods. 50 μg of each digest was subsequently fractionated into 20 fractions using a Waters acquity CSH C18 1.7 μm 1.0 × 15 mm column on a Ultimate 3000 HPLC (Dionex, Sunnyvale, CA, USA) operating at 30 μL/min. Buffer A consisted of 5 mM ammonium bicarbonate and buffer B consisted of 5 mM ammonium bicarbonate, 90% acetonitrile. Proteomics data was searched using MaxQuant (ver. 1.5.6.2).

## Mass spectrometry analysis

Peptides were eluted into 96-well microtiter plates with 20 μL 40% MeCN/0.5% acetic acid followed by 20 μL 60% MeCN/0.5% acetic acid. Peptides were reconstituted in 10 μL 2% MeCN, 0.5% acetic acid, 0.1% TFA after vacuum centrifugation in a speed-vac. Five microliters of the peptide eluate was separated by a linear MeCN gradient for 160 min in a 15-cm fused-silica emitter packed with reversed-phase ReproSil-Pur C18-AQ 1.9 μm resin (Dr. Maisch GmbH) using a nanoflow Easy-nLC system (Thermo Scientific). The LC was connected through a nano-electrospray ion source to the mass spectrometer. We either used a QExactive orbitrap or QExactive HF orbitrap instrument using a top6 higher-energy collisional dissociation (HCD) fragmentation method[50].

## Peptide identification

Raw MS files were processed using the MaxQuant software (ver. 1.6.0.1, Max-Planck Institute of Biochemistry, Martinsried) and Mascot (ver. 2.6.2). HCD-MS/MS spectra were de-isotoped and filtered using the 10 most abundant fragments per 100 $m/z$ range. Peptides were identified by searching all MS/MS spectra against a concatenated forward/reverse target/decoy of the mouse complete proteome (proteome ID: UP000000589). The HCD-MS/MS spectra were searched with Oxidation (M), acetylation (N-term), pyro-Glu (Q/E), amidation (C-term) as variable modifications and with no enzyme specificity. Search parameters were set with precursor ion tolerance of 4.5 p.p.m. and MS/MS tolerance at 0.02 Da. FDR were set at 0.05 at protein level and 0.01 at peptide level. Minimum peptide length was set to seven amino acids and peptides identified with a Mascot score of less than 20 were discarded. All raw data was curated and annotated and stored on an in-house build Omics-manager system. For the MaxQuant search of all tissues combined we used the mouse complete proteome downloaded from Uniprot 28th June 2017. Both proteomics and peptidomics data have been deposited to the ProteomeXchange Consortium[51] via the PRIDE[52] partner repository with the data set identifier PXD022225.

## Data quality assessment and analysis

One rawfile (DB_09 from Sc. Fat) was truncated and removed from subsequent analysis. Data quality plots were made in R-studio (ver. 4.1.0; 2021-05-18). Strain and tissue specificity plots were generated using UMAP library in Python[53] (ver. 3.9.7). Remaining data assessment plots were made with Seaborn (ver. 0.11.2) and Matplotlib (ver. 3.5.1) in Jupyter notebook (ver. 6.4.7). Gene ontology was done using geneontology.org[41]. Kullback–Leibler divergence logo plots were calculated as (ref. 54).

## PPV model

The positive training set used in the predicted-peptide variant (PPV) model consists of known peptides and pro-peptides extracted from Uniprot the 26th January 2017. We added annotated peptides from the SwePep database[33] and the NeuroPep database[34] to construct a curated list of known annotated peptides (Supplementary Data 3). Delayed annotation was determined by extracting known annotated peptides from Uniprot version 2021_02. For PPV predictions in each individual tissue, we used pertinent Mascot files, and a combined MaxQuant search covering that tissue. Output files were merged and combined as follows: If a peptide was exclusively observed in MaxQuant or present in both datasets we used the abundance value provided from Max-Quant. If a peptide was exclusively observed in Mascot, we fitted a transfer function as a first-degree polynomial $\log(y) = a * \log(x) + b$, where the target y was MaxQuant abundance predicted by the x Mascot abundance. The PPV model itself is based on Logistic Regression and written in Scikit-learn (ver. 1.0.2) in python[55] (ver. 3.9.7). When training the logistic (PPV) and comparative non-linear models we used nested 5-fold cross validation to ensure that reported performance metrics are based only on data unseen by the models during training and optimization. For the PPV-assembly model, were trained on an up sampled dataset, and evaluated using nested cross-validation. 14 mass spectrometry features were engineered from the data (Supplemental Note 1). Feature frequency plots were made with Matplotlib (ver. 3.5.1). Calibration curves were made with Seaborn (ver. 0.11.2) in Jupyter (ver. 6.4.7). The PPV source code can be downloaded freely from: https://github.com/jancr/ppv. Supplementary Data 5, 6 contains all PPV predictions and PPV-assembly predictions with a score higher than 0.01.

## Bacterial peptide production

DNA sequences encoding peptides above 35 amino acids in length were fused C-terminally to a 6xHis-SUMO-tag and cloned into pET11d-derived plasmid. The plasmid was transformed into an *E. coli* BL21(DE3)-derived host stain, HNC54, expressing the NucB nuclease from *Staphylococcus aureus* (UniProtKB: A0A447ZB30). To over-express 6xHis-SUMO-fusions, individual clones were inoculated in a 96-well format containing 1.2 mL Terrific Broth containing ampicillin in each well and cultivated at 37 °C with shaking at 800 rpm for 4-5 h. IPTG was added to 0.5 mM final concentration to induce expression and cultivation continued at 37 °C for 16 h. Cells were harvested by centrifugation and resuspended in 100 μL PBS pH 7.4 and freeze-thawed twice. Resulting cell lysate was mixed with 1.2 mL PBS pH 8.0 containing 8 M urea and 10 mM imidazole and centrifuged and the supernatant loaded on a Ni-NTA resin (QIAGEN) column pre-equilibrated with PBS pH 8.0 containing 10 mM imidazole. After washing with 20 bed volumes of PBS buffer, the fusion peptide was eluted with PBS pH 8.0 containing 500 mM imidazole. A PD10 column (GE Healthcare Life Sciences) was used to change the buffer into PBS containing 20 mM imidazole pH 8.0. The eluted fusion-peptide was digested with a SUMO protease at 1:500 (w/w, enzyme/protein) at room temperature overnight. The peptide was recovered in the flow-through fraction using a Ni-NTA column, and purified with Source 30 reverse-phase chromatography (GE Healthcare Life Sciences). Briefly, the Source 30 RPC column was washed with 5 bed volumes of 20 mM ammonium bicarbonate, and washed with 10 bed volumes of 0.5 M arginine at pH 9.0. The peptide was eluted with 40% isopropanol in 20 mM ammonium bicarbonate and the eluate evaporated with SpeedVac (ThermoFisher Scientific) to less than 0.5 mL, and then lyophilized to dryness. The lyophilized product was reconstituted in PBS pH 7.4. Peptides used for in-vivo dosing were checked with respect to purity, molecular mass, and endotoxin levels (<1 EU/kg animal)

using RP-HPLC and Kinetic Turbidimetric LAL testing according to the supplier (Charles River Laboratories).

## Peptide array synthesis

Synthesis were carried out in four 96-well plates using an Intavis Multipep RSi synthesizer using Fmoc based solid phase peptide synthesis (SPPS). For C-terminal amide Fmoc-PAL AM resin from Novabiochem was used and for C-terminal acids the corresponding preloaded TGT resins from Novabiochem were used. Each peptide was synthesized on 5 µmol scale based on resin loading. The synthesis protocol used double deprotection using 20% piperidine + 0.1 M oxyma in DMF and triple couplings using 6 equivalents amino acid/DIC/oxyma 1:1:1 in each coupling and for a total of 3 h coupling time. Cleavage of peptides from resin was carried out using TFA/DTT/TIPS/water 96:2:2:2 (1.2 mL) and the cleaved peptides were collected in corresponding 96-well deep well plates. The volume of TFA was reduced under a stream of nitrogen to 200-300 µL per well. The peptides were precipitated with 1.2 mL ether and filtered and washed in deep-well Solvinert plates with a hydrophilic membrane. The crude peptides were dried using vacuum and redissolved in 500 µL DMSO and collected in 96-well plates. All peptides were analysed by UPLC-MS for assessment of purity. The DMSO stocks were lyophilized, then redissolved in acetonitrile/water 1:1 (1 mL) and then lyophilized again to produce the peptides as white powders.

## Formulation of peptides

Freeze-dried peptides were rehydrated and formulated on 96-well plates. Endotoxin contents of 2 formulations per well plate were measured using LAL assay to rule out bacterial contamination. Rehydration buffer for each peptide was based on its calculated isoelectric point (pI) so that its pH was ≥1 pH unit away from pI. Peptide concentration and purity was checked after rehydration with RP-UPLC using Acquity BEH C18 1.7 µm 2.1 × 30 mm column (Waters) and detection at UV215 nm[56]. Mobile phases A & B consisted of 0.1% TFA in $H_2O$ and 0.1% TFA in 80% $ACN/H_2O$, respectively, and elution was carried out from 20% to 95% phase A at 30 °C. Peptides with <40% purity were discarded, and the rest formulated at 300 µM in 50 mM phosphate at pH 6.0, 7.0 or 8.0, 12 mg/mL propylene glycol and 0.5 mg/mL polysorbate 20 and sterile filtered into glass vials for in vivo studies and plastic Micronics tubes for in vitro studies under LAF. Freeze-thaw (F/T) stability was checked by measuring peptide concentration and purity using RP-UPLC before and after 3 F/T cycles. Formulations were flash-frozen and stored at −80 °C until use. Peptides displaying in vitro or in vivo effects were resynthesized and purified at >90% purity and reformulated using same formulation protocol.

## Peptide administration and blood glucose measurement

Male diabetic BKS.Cg-*Dock7^m*+/+ *Lepr^db*/J (stock no: 000642, Jackson Laboratory, USA) older than 11 weeks and with blood glucose levels higher than 16 mM were selected and allocated to different treatment groups by randomization based on blood glucose. The mice body weight was measured and recorded. Peptide solution was put at room temperature at least 30 min before dosing and dose solution was prefilled into 0.5 mL 29 G insulin syringe (Insulin syringe, BD, USA) according to body weight. Baseline blood glucose was taken before peptide dosing and then an intraperitoneal injection was done to mice with dosing volume at 5 mL/kg per body weight at indicated peptide concentration. Blood glucose was measured at 30 min, 1 h, 2 h, 4 h, 6 h post dosing. In the acute daily peptide dosing study, animals were dosed at morning (~9 am), in the 4 days BID peptide dosing study, animals were dosed in the morning (around 9 am) and after 4 h BG measurement (around 13 pm). Tail vein blood for BG measurement was collected into heparinized glass capillary tubes and 5 µL subsequently mixed with 250 µL of EKF system solution and vortexed to a homogeneous solution. Glucose level was measured using glucose

oxidase method (glucose analyser, BIOSEN 5040, Germany). At termination mice were anesthetized with isoflurane and blood was taken from the orbital puncture and transferred into two EDTA coated tubes and euthanized.

## Surgery and liver perfusion

Perfusion buffers were pre-warmed to 37 °C and gassed with 95:5 $O_2:CO_2$ for approximately 15–30 min prior to surgery. 8 weeks old male Sprague Dawley *Crl:CD(SD)* rats (Charles River) were anesthetized with a mixture of $N_2O/O_2$/isoflurane. Once animals had reached a surgical plane of anaesthesia, a U-shaped incision was made from the rib cage to the lower abdomen through the skin and muscle layer leaving the ribcage and diaphragm intact. The viscera were exposed, and the stomach and intestines were displaced to the right to reveal underlying blood vessels. The hepatic portal vein was isolated and cannulated with a 21G luer adapter connected to prefilled catheter immersed in 250 mL of perfusion buffer 1 (Kreb's Ringer with 20 mM glucose, 120 mM NaCl, 2.8 mM $NaHCO_3$, 20 mM glucose, 1%1 Sol'n C, 5 mM HEPES, 100 µM EGTA, and pH adjusted to 7.4). Perfusion buffer 1 was slowly infused at a rate of ~5 mL/min using a peristaltic pump. The diaphragm was punctured to expose the thoracic cavity and the aorta severed. The flow rate was then slowly and steadily increased to 25 mL/min. A vacuum pump was used to remove perfusion/blood overflow. Perfusion was continued until less than 5 mL of the initial 250 mL volume of perfusion buffer remains. Perfusion was then switched to a perfusion buffer 2 (120 mM NaCl, 2.8 mM $NaHCO_3$, 20 mM glucose, 1%1 Sol'n C, 5 mM HEPES, 1.4 mM $CaCl_2$, adjusted to pH 7.4) with a collagenase blend (7 mg/ml, Liberase TM Research Grade, Roche). Perfusion was continued until less than 5 mL of the 250 mL perfusion buffer remains. The liver was then immediately excised and processed for hepatocyte isolation. At the completion of the procedure the heart was removed to ensure euthanasia.

## Hepatocyte isolation

After perfusion, the rat liver was excised and transferred to sterile dish containing 20 mL wash medium (Gibco #11043023), 5.5 mM glucose, supplemented with 5 mL Penicillin/Streptomycin 1% (Gibco #15140-114), 100 nM dexamethasone, 10% FBS (Ausbian #VS500T), and 1 nM insulin (in house produced), and teared with forceps. Cells were dispersed by gently aspiration and filtered through a 100 µm nylon filter into a 50 mL tube. Cells were pelleted at 100 g at 4 °C for 3 min. Supernatant was decanted and cell pellets dispersed by aspirating gently with a bore pipet in 10 mL wash medium. Washing and resuspension steps was repeated for 3 times. Isolated hepatocytes were suspended in 20 mL Basal media (as wash medium but with 4% FBS). Cell number and viability were counted with a haemocytometer. Hepatocytes were diluted and plated into collagen-coated wells (30,000/well) and cultured in basal media for 4 h, then changed to starvation media (as wash medium with 0.1% FBS) overnight.

## PEPCK qPCR

Primary hepatocytes were prepared as described above. Cells were treated with serial diluted peptide, 10 nM insulin and 10 nM glucagon (both controls in house produced) for 4 h. After treatment, cells were lysed, and RT-PCR was performed with Cells-to-CT Kit (Ambion #4402955) according to manufacturer's recommendation. In brief, cells were lysed with 25 µL lysis buffer with shaking at 750 rpm for 10 min before 2.5 µL stop solution was added. Reverse transcription was performed in 384 well-PCR plate. qPCR analysis was performed with Applied Biosystems ViiA7 instrument (Life technologies). G6pc and Pck1 expression in H4IIE cells were measured and normalized to Rplp0. PEPCK, G6PC and Rplp0 qPCR primers were purchased from ThermoFisher; PCK1 TaqMan Gene Expression Assays (Rn01529014 m1), G6PC TaqMan Gene Expression Assays (Rn00689876 m1), RPLP0 TaqMan Gene Expression Assays (Rn03302271 gH).

## Glucose uptake assay

3T3-L1 MBX fibroblasts (ATCC, #CRL3242) were seeded at 20,000 cells per well in collagen coated 96-plates (Corning #356650) in DMEM (Gibco #10569) with 10% FBS (Ausbian #VS500T). Cells were differentiated into adipocytes for 4 days with 3% FBS, 1 μg/mL human insulin (in-house produced), 0.5 mM isobutylxanthine (Sigma #I5879), 1 μM dexamethasone (Sigma #D4902) and 2 μM Rosiglitazone (Sigma #R2408), followed by 3% FBS and 1 μg/mL insulin for another 7 days, then maintained in 3% FBS for 8–11 days. After differentiation, cells were treated with diluted peptide and 100 nM insulin as positive control overnight in DMEM low glucose medium (Gibco #A1443001). Glucose uptake was measured by Glucose Uptake-Glo™ assay kit (Promega #J1343) according to the manufacturer's recommendation. Briefly, cells were incubated with 0.5 mM 2DG in PBS for 10 min at 25 °C, before stop buffer and neutralization buffer were added sequentially. 2DG6P detection reagent were added to cell samples and incubated for 1 h before luminescence measurement on Envision (PerkinElmer). Data was analyzed with GraphPad Prism v7.

## Glucose stimulated insulin secretion assay

The INS1E luciferase cells developed in the lab of Claes Wollheim, was transfected with the proinsulin-Luciferase reporter construct[57]. INS1E Luciferase clone21 cells were seeded in pre-coated 96 plates at 50,000 cells/well and cultured for 3 days in RPMI1640 medium with GlutaMAX (Gibco #61870), 1% Penicillin/Streptomycin (Invitrogen #15140-122), 50 μM mercaptoethanol (Sigma #M3148-25ml), 10% fetal bovine serum (GIBICO #10100-147), 100 mM sodium pyruvate (GIBICO #11360), 1 M HEPES (GIBICO #15630). cells were washed with KRBH buffer (1x Krebs-Ringer buffer) supplemented with 10 mM HEPES (#15630-080), 1% GlutaMAX (#35030-061), 24 mM NaHCO$_3$ (#25080-094), 0.2% BSA (SIGMA #B2064), 1% Penicillin/Streptomycin (#15140-122) and adjusted to pH 7.4. 100 μL KRBH with 0.5 mM glucose were added and incubated at 37 °C for 2 h. Cells were subsequently washed with KRBH buffer once before 100 μL KRBH with diluted peptides in 16.7 mM glucose were added at 37 °C for 30 min. After treatment the cells were centrifuged at 100 $g$ for 2 min. 50 μL supernatant was transferred to a new 96-well plate. Nano-Glo luciferase assay (Promega #N1120) were performed according to manufacturer's recommendation. In brief, Nano-Glo luciferase buffer was mixed with substrate at 50:1 ratio, before the mixture was added to cell supernatant at a 1:1 ratio. The mixture was incubated at room temperature for 10 min before luminescence measurement on Envision (PerkinElmer). Data was analyzed with GraphPad Prism v7.

## Hydrodynamic gene delivery

CHGB plasmids and control plasmids were manufactured by Genscript (Nanjing, China) at transfection grade with endotoxin <25 EU/mg, and all plasmids were diluted in saline to a concentration of 20 μg/mL, 2.5 mL of pre-warmed dose solutions with 50 μg construct or 10 μg as positive control were carefully injected into conscious mouse through tail vein within 5 s using a 25 G syringe. Plasma was collected on termination day.

## Plasma insulin measurement

We employed Luminescence Oxygen Channeling Immunoassay (LOCI). 4 μL sample/calibrator/control was applied together with 7 μL assay buffer in 384-well LOCI plates coated with mAb HUI018 (5 μg/mL) conjugated acceptor-beads corresponding to 35 μL/well. The assay plate was shaken for 1 h at room temperature, and after wash 10 μL of biotinylated guinea pig anti-mouse insulin Ab 4077 (6 μg/mL) was added to each well (10 μL/well). Both the mAb HUI018 and pAb 4077 are in-housed produced. The assay plate was again shaken for 1 h at 22 °C. After washing, 10 μL streptavidin coated donor beads (67 μg/mL) were added to each well and incubated for 60 min at 22 °C. Plates were read in an Envision plate reader with a filter having a bandwidth of

520–645 nm after excitation by a 680 nm laser. The total measurement time per well is 210 ms including a 70 ms excitation time. The lower limit of quantification is 20 pM.

## Plasma β−3-hydroxybutyrate (3-HB) measurement

3-HB were measured using a Cyclic Enzymatic Method and an auto biochemical analyzer Cobas C501 with reagents (Cat no. 41773501/41373601/41273791) purchased from Wako Chemicals GmbH, D-41468 Neuss.

## Plasma triglycerides measurement

Triglycerides were measured by auto biochemical analyzer Cobas C501 with reagents (Cat no. 20767107322) purchased from Roche Diagnostics GmbH, D-68298 Mannheim.

## Reporting summary

Further information on research design is available in the Nature Research Reporting Summary linked to this article.

## Data availability

Source data are provided with this paper as Supplementary Data files 1–6 and in the Source Data file. The raw mass spectrometry data and processed search files have been deposited at the ProteomeXchange Consortium via the PRIDE partner repository with the data set identifier PXD022225. Public peptide databases such as SwePep (http://www.swepep.org/), Uniprot (https://www.uniprot.org/) and NeuroPep (http://isyslab.info/NeuroPep/) were used for Supplementary Data 3.

## Code availability

The PPV code is available in the GitHub repository at https://github.com/jancr/ppv, released under the MIT licence (https://zenodo.org/record/7140868#.Y0EvSNhBw2w). All plots can be regenerated from the Jupyter notebook: https://github.com/jancr/ppv/blob/master/notebooks/manuscript_figures.md.

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

## Acknowledgements

We thank Dr. L.G. Grunnet, Dr. C. Rescan, and I.P. Eliasen for Islet and GSIS assay support. Dr. T. Frogne is acknowledged for the creation of the INS1E luciferase clone21 cell line. Dr. J.O. Samuelsson and Professor J. Cox for data logistics and search engine support. Y.T. Lam, A.K. Hansen, and M.B. Larsen are thanked for technical MS support. Y. Gan is thanked for *E. coli* expression support. Y. Wu, X. Yang, X. Sun, Y. Dai, Y. LV are all thanked for in-vitro support, and W. Yin, and H. Sun for in-vivo support. B. Wang and J. Jin are thanked for plasma analysis support. J. Yang and K.A. Richard are thanked for invaluable data support. The Novo Nordisk Foundation (Grant agreement NNF14CC0001) supports J.V. Olsen's work at the Novo Nordisk Foundation Center for Protein Research.

## Author contributions

C.T.M. performed mass spectrometry and collected all data. J.C.R., S.K.K., and F.G.T did the bioinformatics and model work. Z.W., Y.T. designed and executed in-vitro screening and target validation; G.M., Q.J., X.Z., J.J. designed and executed in-vivo screening and target validation; F.Z., D.H., and X.C. contributed to screening strategy and data interpretation. C.J. synthesized all chemical peptides. P.H. formulated all peptides. U.L. supervised and designed computational strategies and workflows. B.W. and X.Z. expressed peptides in *E. coli*. C.D.K. and J.V.O. assisted in data interpretation and manuscript preparation. C.T.M., S.T., J.N-J., S.T.B., and M.G. initiated and planned peptidomics activities. M.G and U.L contributed equally. C.T.M. and U.L. wrote the manuscript.

## Competing interests
