## [Peer Review File · Nature Communications]

REVIEWER COMMENTS

Reviewer #1 (Remarks to the Author):

This manuscript describes the generation of a database of peptides identified by mass spectrometry from a range of mouse tissues, together with the development of a prediction programme, potentially able to identify bioactive peptides from mass spectrometry data (the needle in the haystack). The programme was trained against peptidomics data from a variety of tissues which included known bioactive peptides, and then used against a much larger dataset. It was ultimately capable of picking up the peptides against which it was trained, as well as others that had either been mislabelled, or that were not in the training dataset.

Overall, the programme will certainly be of value to researchers in the field if made freely available, as the authors are correct in stating that it is laborious, and near impossible, to identify potentially bioactive peptides by manually sifting through mass spec peptidomic databases that contain degradation fragments and non-active products of proteolytic cleavage reactions.

I am less convinced, however, that the authors were successful in identifying new bioactive peptides using this approach. They describe a few peptides identified by the algorithm that were taken through to in vivo and in vitro studies, but these were only effective at very high concentrations in a limited number of experiments, and the in vivo findings could not be backed up by in vitro mechanism of action studies.

Specific points:

1. The authors have referenced other bioactive peptide prediction software resources on page 4 – for example peptide ranker by Mooney et al. How do the peptides that were selected for administration rank on these other peptide prediction packages compared to PPV?
2. The proteomics part of the paper should arguably be cut, as it isn't clear how it adds to the manuscript. The authors say that the proteomics data back up the strain separation for figure 2B compared with figure 2D, however their explanation for including it to confirm the peptides aren't derived from degradation of large proteins is not only confusing, but also a very weak argument. I don't see the need for including this proteomics data set in a paper discussing the identification of bioactive peptides using mass spectrometry.
3. Figure 3b is particularly difficult to follow and interpret. Whilst this is a potential way to show degradation, using just peptide spectral matches doesn't necessarily show the whole picture. Likely the authors have already attempted different ways of displaying these results. However, one alternative would be to compare the peak area or intensity of each matched peptide relative to the parent peptide. This will indicate the relative levels of degradation from each method, as well as which peptides might be generated at sufficient quantities to have bioactive potential when diluted into the bloodstream.

If intensity data aren't available from the other data sets, another way of showing the data would be to plot the number of peptides identified for each protein as a function of the parent protein molecular weight. The majority of bioactive peptides come from low molecular weight precursor proteins (bar some of the non-canonically cleaved peptides) therefore this approach would also highlight the challenges in differentiating peptides that are generated from unwanted protein degradation and those that are truly real bioactive peptides.

4. A small subset of pancreatic sample data files were downloaded and put through a peptidomics search by this reviewer. Surprisingly, there were no A or B chain matches against insulin in the data. I couldn't see reduction and alkylation being discussed in the method section (information on page 18 doesn't include carbamidomethylation). If reduction and alkylation isn't performed, then a large subsection of bioactive peptides could be missed. The supplemental data set (supplementary figure 2) shows data where peptides were reduced and alkylated to check the recovery of the MWCO devices – but it wasn't used in the main experimental dataset? Please discuss your reasoning for not performing reduction and alkylation on the main dataset.

5. The pancreas peptidomics data also had a low signal for the glucagon peptide in a fair number of the samples. I would have expected a much higher signal in the pancreas for glucagon, as this is one of the main peptide products from the pancreas. Coincidentally, there was a significant amount of degradation of the glucagon peptide as well as the other peptides from the proglucagon gene. Although this doesn't reduce the validity of the algorithm for identifying peptides, it does highlight the issue the need to optimise tissue processing to preserve native peptides, which should be discussed.

6. Can the authors include figures showing spectral data and annotated product ion spectra for the novel insulin peptides – this would give more confidence in the identification of the splice variants that also contain the C-terminal amidation modification.

7. The authors describe the ability to detect Manserin in the whole pancreatic tissue (page 7), but not in islet supernatants, which they use as an argument for analysing tissue peptidomics to identify peptides for which the secretory conditions are unknown. However, this approach will also identify a large number of peptides that are never secreted, and are effectively false positives. The lack of detection of manserin in islet supernatants is likely due to the sensitivity and recovery of the secretory peptidomics being sub-optimal. This limitation should be discussed, together with some discussion of the authors' opinion about the likelihood that individual peptides identified by the algorithm, but deriving from large cytosolic proteins, will turn out to be genuine bioactive peptides rather than false positives.

8. With the atypical C-terminal amidation findings for peptides, how certain are the authors that these are real and not instrumental artifacts? For larger peptides, the precursor ion that is selected for fragmentation can sometimes be erroneous – giving an apparent ~1 Da mass reduction. If the subsequent fragmentation contains mainly b-ions – which is quite common in peptidomics, then this might also give good peptide sequence matches with good mass accuracy, and therefore an incorrect C-terminal amidation assignment. Have these atypical modifications been confirmed by manual curation of the precursor ion C13 cluster and the respective product ion spectra?

9. The experiments in suppl fig 6 claim to validate the importance of C-terminal amidation for neuromedin C. However, the non-amidated version was only tested at a lower concentration in vivo, and was not tested in the insulin secretion experiments. To conclude that amidation is important, the amidated and non-amidated peptides need to be compared at the same concentrations and doses. Notably, stimulation of insulin secretion in vitro was only observed at very high concentrations (10 and 20 microM), arguing against this targeting a specific high affinity receptor.

Reviewer #2 (Remarks to the Author):

This is an interesting study and Madsen et al. have conducted a comprehensive analysis of the mouse peptidome across seven tissues in four different mouse strains. A machine learning model was trained by using this dataset, which included the derived features of positional and encode aspects, observed modifications, physiochemical properties, etc. Many candidates of bioactive peptides were predicted, which showed the features similar to those of the known bioactive peptides. They also performed experimental validation using two predicted peptides in diabetes-related models. Overall, this paper is generally well written and structured. They offer a useful tool for predicting bioactive peptides and identified two peptides with potential function. The reviewer has some questions and concerns as shown below.

Major issues:

1. No category on bioactivity is specified. The authors have demonstrated the utility and efficacy of the PPV model by predicting thousands of high-scoring bioactive peptide candidates. However, bioactivity is a very broad term, and the PPV model is not able to predict what functional roles these peptide candidates possibly play, and what biological and physiological processes they modulate, such as antimicrobial activity, neuromodulation, endocrine regulation, etc. More importantly, the authors need to define the term of “bioactivity” or “bioactive peptide”, which is directly related to the value and significance of the machine-learning model and the predicted data resource of bioactive peptides. In other words, if we have a predicted bioactive peptide candidate, we do not know how to examine its function and to obtain certain phenotype in vivo or in vitro.

2. The reviewer also concern that “in-silico validation” uses the traditional features for neuropeptide prediction. In Figure 5a and 5b, the authors observed the two features, i.e., di-basic cleavage motifs and secreted proteins. Those are the molecular features routinely used for prediction of neuropeptide candidates. If the precursor protein has been considered in the mass spectrometry derived feature (page 8 of Technical Note), it is not surprising that the two features were seen in the predicted candidates.

Minor issues:

1. In the abstract (page 2), it is better to address the number of the peptides predicted in this study to indicate the size of this data resource, and also highlight these validated peptides with biological functions in this work.
2. In the line 7 of page 6, why “our data indeed reflects the in vivo situation”? Please explain.
3. In the line 21 of page 6, please indicate the criterion for “perfect sequence matches”, such as Mascot score, sequential fragment ion number, etc.
4. In the line 15 of page 7, is it possible that the Manserin peptide is secreted, but is below the detection limit?
5. In the line 5 of page 11, please show the MS/MS spectrum of this novel 22-amino acid peptide. To achieve a confident validation, it is better to fragment a chemically synthetic peptide and show the mirror MS/MS spectrum for a comparison.
6. In the line 10 of page 12, we can not exclude the possibility that the amidation is formed on the side chains of amino acids through glutamic acid to glutamine, and aspartic acid to asparagine, instead of C-terminal amidation.
7. In line 18 of page 12, the authors selected these peptide candidates for functional validation based on gene ontology annotations. Please explain the underlying mechanism. Is it possible to predict the potential functions of these predicted bioactive peptides on that?
8. In line 17 of page 16, some mice are 15 weeks old and others are 26 weeks. Is there an age difference in the datasets?
9. In Figure 1b, the Mascot individually identified 26411 peptides, and MaxQuant is 30221. It is interesting to summarize the Mascot score and Andromeda score on the three areas in the Venn diagram to see if the individually identified peptides have a low identification confidence.
10. In Figure 2a and b, UMAP analysis was used for tissue and strain specificities, while PCA analysis was performed at protein level in Figure 2d. Why different analysis was used. How about the PCA analysis results on tissue and strain specificities?
11. There is no x-axis label in the left and central panels of Figure 4a.
12. In Figure 8b and c, there should have negative controls in the two experiments by injecting non-functional peptides.

Reviewer #3 (Remarks to the Author):

A hot topic in peptide research is the identification of bioactive peptides. The authors of this study carried out a large-scale peptidomic analysis of seven different mouse tissues from four different strains. Based on an imbalanced dataset (207 positive and 217695 negative samples), the authors developed a Bayesian logistic regression model. The proposed framework is capable of identifying candidate peptides, which has been confirmed through experimental approaches. Although the authors employed a simple approach, the model still needs to be improved. My concerns are as follows:

1. It is unclear why the authors chose Bayesian logistic regression for model building. Moreover, there are several ways to handle the imbalanced dataset problem. I recommend using undersampling techniques or SMOTE to address this imbalanced problem, and then applying a set of conventional classifiers and evaluating their performance. Compare these results with the proposed approach, which will give a clear picture to the readers.
2. The positive sample length ranges from 7 to 42 amino acids. On the other hand, the negative sample length ranges from 7 to 79 amino acids. What is the rationale for considering more than 42 amino acids in the negative samples?
3. To examine the robustness of the developed model, the author should use a portion of the original data (10% SHOULD NOT be used in training) and check the transferrability of the developed model (trained with 90% of the data).

Point by point response to the reviewers.

General comments:

We would like to thank both the reviewers and the editorial team for critical reading of the manuscript and for their constructive comments that we feel have helped us substantially improve our study and manuscript.

We have put special emphasis on four main aspects that emerged from the feedback we received.

First, concerns were raised about the entire machine learning pipeline including the validity of the performance evaluation, model selection, potential issues with the unbalanced dataset, comparisons to sequence-based prediction methods, potential bias from differences in length distribution within the training data, etc. To address these concerns, we have re-designed and re-trained the entire machine learning framework, implementing the following changes:

- We now use nested 5-fold cross validation to ensure that reported performance metrics are based only on data unseen by the models during training and optimization and that the different splits of the data do not contain overlapping examples that would result in misleading performance metrics.
- We included the comparison of different modelling framework (both linear and non-linear models) to explain and motivate the final choice of model for the PPV method, including the regularization techniques suggested by reviewer 3. The overall picture remains the same as in the original manuscript, but we now show the motivation for basing the PPV model on logistic regression.
- We have simplified the PPV modelling framework (without losing predictive performance) to make it clearer to readers how it works, what it predicts and what drives the predictive power.
- The simplification involves not including the amino-acid sequence derived features (which we found to add very little to the performance) and this has the additionally benefit of making the PPV model fully orthogonal to alternative sequence-composition based methods such as PeptideRanker. As suggested by reviewers, the revised manuscript makes a direct head-to-head comparison which demonstrates that the PPV model (now based only on MS data derived features) clearly outperforms PeptideRanker (which uses only the amino acid sequence) on the task of identifying known bioactive peptides from the peptidomics data.
- We illustrate how the two orthogonal methods can be used in combination to provide additional support to the output of the PPV model.
- We address the question of imbalance in the training set (raised by reviewer 3) by showing that this has little to no influence on the relative ranking of the predicted peptides, meaning that the final lists of predicted peptides are essentially the same when down-sampling the negative set by a factor of 10x during training.

Second, we realize that we were not sufficiently clear about what exactly our PPV method is predicting and how that, combined with other data, can be used to identify promising candidate bioactive peptides and guide experimental validation experiments. We have therefore re-designed the prediction model itself and adjusted our descriptions to make it much clearer to the reader exactly what the PPV model predicts, how it works, and how we believe the output can best be utilized towards the goal of identifying new bioactive peptides.

Third, concerns were raised about aspects of our predicted peptides and the in-silico and experimental validation. To strengthen this part, we have added a number of additional analysis to prove that the MS findings are indeed solid and correct as well as added additional experimental data for our validation

examples, including negative controls to prove that injecting other (non-functional) peptide sequences at high doses does not produce a response in our assays and models.

Forth, we have rewritten the entire discussion section at the end of the manuscript to highlight the limitations and potential points of improvement across all levels of our study (including tissue handling, MS analysis, machine learning, and the utility of the predictions). The reviewer feedback clearly revealed to us that these aspects were too poorly addressed and explained in the original manuscript.

Below, we provide point-by-point responses to the individual reviewer comments (in reverse order, starting with reviewer 3).

Reviewer #3 (Remarks to the Author):

A hot topic in peptide research is the identification of bioactive peptides. The authors of this study carried out a large-scale peptidomic analysis of seven different mouse tissues from four different strains. Based on an imbalanced dataset (207 positive and 217695 negative samples), the authors developed a Bayesian logistic regression model. The proposed framework is capable of identifying candidate peptides, which has been confirmed through experimental approaches.

We thank the reviewer for appreciating the methodology developed for identifying new potentially bioactive peptides, in addition to the constructive feedback in relation to the machine learning approach.

Although the authors employed a simple approach, the model still needs to be improved. My concerns are as follows:

1. It is unclear why the authors chose Bayesian logistic regression for model building. Moreover, there are several ways to handle the imbalance dataset problem. I recommend using undersampling techniques or SMOTE to address this imbalanced problem, and then applying a set of conventional classifiers and evaluating their performance. Compare these results with the proposed approach, which will give a clear picture to the readers.

We agree complete with the reviewer that leaving out the model comparison and selection part was a clear mistake on our part. We now include a comparison of different modelling frameworks in Fig. 4c and Supplementary Figure. 7a-c to support our model choice:

It demonstrates that logistic regression is still the best method, particularly when evaluating the ability of the models to enrich for known peptides among the top300 predictions. It outperforms the two non-linear comparators, Random Forest and Support Vector Machine (SVM), and neither Elastic net nor SMOTE (suggested by the reviewer) provided improvements over simple logistic regression.

All models were re-trained using nested 5-fold cross-validation with careful consideration to the way the data was split into the five bins to ensure an unbiased fair assessment of model performance (See Technical Note and later comments to reviewer feedback).

Several of the models show highly similar AUCs on the full data set (ca. 200.000 peptide training examples). But, since the purpose of the PPV method is to rank potential peptide candidates to highlight only the most

promising ones (among tens of thousands) for further investigation, we base our model selection on the performance on the top300 predicted peptides. Logistic regression therefore remains our model of choice, due to its good performance and explainability (also see updated framework illustration in Fig 4a).

One of the main motivations for picking Bayesian logistic regression in the original manuscript was to provide the readers with an estimate of the relative importance and directionality of the input features, including the level of uncertainty in the estimation of each input feature weight.

We now instead obtain a similar result for simple logistic regression via the nested 5-fold cross-validation approach which generates a total of 20 models, each with a set of regression coefficients (weights) that allow us to visualize the distribution of these for each input feature (Fig 4b).

The Bayesian version is both harder to explain and computationally more expensive to compute, and we therefore picked simple logistic regression as the basis for the PPV model instead. At its core, it is essentially the same model. We thank the reviewer for pointing out the shortcoming in our original manuscript.

Since our goal with the PPV model is ranking of peptides, the imbalance in the training data that the reviewer points to (few positive examples vs many negative examples) is not a problem per se. To illustrate this point, we included a new analysis (see Technical note) where we compare the prediction output of the PPV model trained using the full set of negative examples versus only using 10% of the negative examples (see figure below).

The under-sampling (using only 10% of the negatives) shifts all the absolute prediction scores upwards, but the relative ranking and the individual distance between peptides remains the same. This is illustrated also with a venn-diagram showing that about 90% of the peptides on the top300 lists from the two models (10% vs full) are the same. The remaining difference is likely due to minor score differences resulting from reducing the training set. Imbalance in the training examples is therefore not an issue when the goal is to obtain a relative ranking to find the most convincing looking peptide candidates across a large data set.

The mathematical explanation is that down-sampling of the negative set simply shifts the prior probability estimate of a peptide being real (a positive); equivalent to shifting the entire scale, as shown in the figure. In a hypothetical fully balanced training scenario (under-sampling to equally many positive and negative examples), the prior probability of a randomly selected peptide from the training set would be 50% which would cause the model to assign very high probability scores to tens of thousands of peptides. We therefore find the model trained on the full data set to be the most fair and accurate estimate of the actual probability of a predicted peptide being real (and potentially bioactive). In fact, the estimates are probably too conservative (too low) because we assume during training that everything not labelled as positive is negative.

As a result of the re-design and re-training, the following figures in the manuscripts has changed; Fig. 4a-c, Fig. 5a-e, Table 1, Fig. 7a,b, as well as the following supplementary figures; Supp. Fig.7a-c, Supp. Fig. 9, Supp. Fig. 16, Supplemental table. 5, and Supplemental table. 6.

2. The positive sample length ranges from 7 to 42 amino acids. On the other hand, the negative sample length ranges from 7 to 79 amino acids. What is the rationale for considering more than 42 amino acids in the negative samples?

We thank the reviewer for noting that differences in length distribution could affect the prediction output. In the new training setup, we therefore only train on sequences up to 42 amino acids in length to mitigate any difference in length distribution between the positive and negative sets.

As part of our re-design and re-training, we decided to omit the amino acid sequence derived features, since they added very little to the model performance and possibly blurred the message that patterns in the MS peptidomics data is the real driver of the prediction power. We additionally left out features which encoded the percentage of the parent protein covered by observed peptides, as this feature could contain information about the presence or absence of a signal peptide (signal peptides are usually not observed in the MS-data because they get cleaves off) and thereby potentially leak information about the secretion status of the precursor protein to the method. Features based on raw occurrence counts were also dropped because they could assume extreme values for a small subset of peptides.

The final set of input features are therefore largely – albeit not entirely - length independent ones (see Technical note) and given the way the PPV model works, we suspected that it would also work relatively well for sequences longer than those it was trained on. This is a relevant question because we know from literature (and our gold standard set from peptide databases) that bioactive peptides longer than 42 amino acids certainly do exist.

We therefore performed a test to explore how a model trained on shorter peptides (7-30 aa) would perform on the longer (>30 aa) known peptides that it had never seen before:

As can be seen from the figure, the model trained on only shorter peptides assigns lower probability scores to the longer known peptides (red dots), indicating a slight bias away from predicting longer peptides. To put it differently, the PPV model framework is overly conservative in its predictions on sequences longer than the ones it was trained on which means that it can confidently be used to predict on longer sequences.

We utilized the final PPV model (trained on peptides 7-42 aa) to predict on all observed sequences which results in a total of 3199 peptides with a PPV > 0.01. 149 (<5%) of these range from 42 to 67 amino acids (Supplementary table 5). Among the top200 predictions, only 11 peptides are longer than 42 with the longest one being 51.

3. To examine the robustness of the developed model, the author should use a portion of the original data (10% SHOULD NOT be used in training) and check the transferrability of the developed model (trained with 90% of the data).

We agree that using the entire data set to train the model could lead to overestimation of the performance on unseen data and are grateful to the reviewer for critiquing this aspect of our original manuscript.

As mentioned above, we now split the entire data set into five independent sets in a way that ensures that occurrences of the same peptide are always in the same sub-set and that the proportion of amidated known peptides (positive examples) is similar across all sets. We refer to the Technical Note for exact details on the splitting procedure.

We then re-trained, optimized and tested models using a 5-fold cross-validation scheme as illustrated in the figure below (included in Technical Note).

Each model is trained on 3/5 of the data and validated/optimized on a 4th set (e.g. parameter tuning). The 5th set is held out entirely and only used at the very last step to measure the predictive performance of that model. The resulting method is thus an ensemble of 20 different models which carries the additional benefit of allowing us to plot the distribution of weights for each (Fig 4b).

The performance metrics are computed for each model (Random Forest, SVM, logistic regression, etc.) by summarizing the prediction scores for each peptide from only those four models which have never seen this example before (not in training or validation set). That way, all prediction scores reported in graphs,

figures and tables in the manuscript originate from an ensemble of models that have never seen the data for that particular peptide before.

We really feel that taking this more rigorous approach has significantly improved our manuscript and thank the reviewers for helping us realize that we had not provided sufficient proof to rule out potential overfitting of the data. In this regards we note that the splitting, cross-validation, re-training, re-evaluation of model performance, and final prediction output reassuringly lead to similar results as those reported in the original manuscript. Logistic regression remains the best performing model, the input features driving the predictive performance are the same, and the peptides that end up high on the scoring lists are also the same. This indicates that the original model probably was not overfitting the data to any significant degree, if at all, and that PPV modelling framework is robust and generalizes well to unseen data.

Reviewer #2 (Remarks to the Author):

This is an interesting study and Madsen et al. have conducted a comprehensive analysis of the mouse peptidome across seven tissues in four different mouse strains. A machine learning model was trained by using this dataset, which included the derived features of positional and encode aspects, observed modifications, physiochemical properties, etc. Many candidates of bioactive peptides were predicted, which showed the features similar to those of the known bioactive peptides. They also performed experimental validation using two predicted peptides in diabetes-related models. Overall, this paper is generally well written and structured. They offer a useful tool for predicting bioactive peptides and identified two peptides with potential function. The reviewer has some questions and concerns as shown below.

We thank the reviewer for appreciating the novelty and usefulness of the developed model that can cope with the scale and complexity of tissue peptidomics data. We are pleased the reviewer find the paper well written and structured.

Major issues:

1. No category on bioactivity is specified. The authors have demonstrated the utility and efficacy of the PPV model by predicting thousands of high-scoring bioactive peptide candidates. However, bioactivity is a very broad term, and the PPV model is not able to predict what functional roles these peptide candidates possibly play, and what biological and physiological processes they modulate, such as antimicrobial activity, neuromodulation, endocrine regulation, etc. More importantly, the authors need to define the term of “bioactivity” or “bioactive peptide”, which is directly related to the value and significance of the machine-learning model and the predicted data resource of bioactive peptides. In other words, if we have a predicted bioactive peptide candidate, we do not know how to examine its function and to obtain certain phenotype in vivo or in vitro.

These are very valid and important points, and we fully agree with the reviewer that it is notoriously difficult to predict the exact biological function of a peptide, even when one knows that it exist in-vivo.

Our PPV predictor is in essence a ranking tool which identifies peptides that look similar (in mass spectrometry feature space) to the set of known annotated peptides, most of which are known to be bioactive. In that way it “indirectly” identifies bioactive peptides. We acknowledge that it might not have been sufficiently clear in our original manuscript exactly what the PPV model predicts, and how the output is best put to use for guiding further experimental follow-up.

In relation to the first point, we now only use input features derived from the patterns observed in the MS data to make it clear to the readers and reviewers that the PPV model is trained to identify peptides which manifest in the MS data with patterns (including shorter, longer and overlapping fragments) that resemble those of the set of known annotated peptides. The class being predicted is thus “similar to known annotated peptides” and since the vast majority of these are known to be bioactive, we make the argument that our approach is suited for identifying novel bioactive peptides, although we acknowledge that it does not predicts bioactivity as such.

Methods like PeptideRanker are also trained to recognize peptides that resemble the already known ones, but such tools instead rely on the amino acid sequence of the peptides as the only input. The two methods are essentially trying to predict the same thing (same class definition) but as we now show in the revised manuscript, the PPV model does a much better job at identifying the known peptides than PeptideRanker does, even when the latter method predict directly on its own training examples (which it may have been

overtrained on). We therefore consider our approach (MS tissue peptidomics analyzed with the PPV model) to be the state-of-the-art method for identifying novel peptides, many of which are likely bioactive.

We have tried to make this point come across much clearer in the revised manuscript, as detailed below.

One of the changes which will hopefully make the machine learning part more understandable to readers is the decision to leave out the sequence-derived features in our re-trained PPV model (they added very little predictive power anyway) such that it only looks at the MS data. This is simpler and reflects that it is the signals in the MS data that is the main driver of the predictions.

The simplification in the input feature space furthermore makes PPV completely orthogonal to sequence-based methods like PeptideRanker and allows for a direct head-to-head comparison (Fig 4c), which indicates that the MS data contains more signal than the amino acid composition towards identifying “real peptides” that are stably formed in-vivo and of which most are probably bioactive. We believe that this has strengthened the significance of our study and methodology.

We also now demonstrate the power of combining the two methods in a new analysis (Fig 5e) and show that the PeptideRanker output provides additional support for some of the exciting, predicted candidate peptides that we highlighted in the original manuscript (e.g. the insulin D-peptide and the tpi1-derived peptide).

The other issue that the reviewer raises here is how one then moves from a predicted and potentially bioactive peptide to making educated guesses as to the biological function of the candidate peptides which can guide experimental validation and mode-of-action studies. We openly admit that we have not been able to come up with a perfect way of doing this, and that we are not aware of any method or approach that can reliably predict the function of a peptide.

Most likely, it is simply infeasible to predict the precise mode-of-action of a peptide from the data available. Firstly, we know that even small changes like enzymatic cleavage of a few amino acids is enough to activate or inactivate a peptide (e.g. GLP1 and angiotensin) and that highly similar peptides may bind to different receptors (GLP1 vs GIP). It is thus an extremely difficult prediction problem to predict the exact biological function from sequence composition, MS-data, genetic variation, gene expression, etc. The relatively small number of known peptides (only about 300 are known in humans) also means that if one starts to further sub-divide these into more fine-grained functional groups, the amount of positive training examples quickly become too small to train a robust reliable model. Methods attempting to do so have been published, such as the MultiPep prediction tool by Grønning et al., but we find the categories predicted (“neuropeptide”) to be of limited utility in guiding assay selection.

To bridge the gap from model predictions to selection of screening/validation assays or animal models, we have expanded on the part where we explain how additional information relating to the gene or protein precursor can potentially be used to home in on the potential function of the peptide. We have also rephrased the part where we explain our own choice of models, assays and peptides to test.

2. The reviewer also concern that “in-silico validation” uses the traditional features for neuropeptide prediction. In Figure 5a and 5b, the authors observed the two features, i.e., di-basic cleavage motifs and secreted proteins. Those are the molecular features routinely used for prediction of neuropeptide candidates. If the precursor protein has been considered in the mass spectrometry derived feature (page 8 of Technical Note), it is not surprising that the two features were seen in the predicted candidates.

We fully agree with the reviewer that the analysis showing enrichment of di-basic motifs in the flanking regions of the high scoring (novel) peptides and the enrichment of peptides originating from secreted proteins would indeed be flawed if information was leaked to the PPV model via the input features. We can, however, assure the reviewer that this is not the case in either the revised or the original work. This comment was nonetheless very helpful as it illustrated the need to be clearer about which input features the model is based on and that these do not contain information – directly or indirectly – on the two phenomena that we later use as “in-silico” validation.

None of the MS-derived features used as input to the PPV model contain any information about the amino acid composition or sequence of the peptide itself or its flanking regions. Since we exclude all the known peptides (incl. all the neuropeptides) from this analysis, the observed enrichment of di-basic motifs is entirely driven by the newly predicted peptides and therefore serves as independent proof that our method systematically identifies peptides (e.g. in brain) that share the well known cleavage motifs of the hormone convertases active in these tissues.

Figure 5a

Figure 5b

The amino acid derived features used in the original manuscript may have contributed to the confusion and lack of clarity here. As mentioned earlier, we therefore removed these entirely to get a cleaner, more understandable model (without sacrificing predictive performance).

When it comes to the secretion status of the parent protein precursor, there may actually have been – at least theoretically – a minor issue with one particular feature which encoded the percentage of the protein precursor covered by observed peptides. This feature (MS Frequency Coverage) could in theory leak information about the presence or absence of a signal peptide and we therefore did not include it in the new model. This feature was not among the main drivers of predictive power and removing it had little influence of the final output. As shown in the updated Figure 5b (see above), the new PPV method still enriches for secreted peptides in a matter that scales linearly with the prediction scores (highest proportion of secreted proteins among the top200 predicted peptides).

As we do in the manuscript, we would also here like to highlight the “in-silico” validation evidence represented by “mis-labelled” known peptides (like NERP-1) that the method correctly identifies, even though they were labelled as negatives during training (due to delayed annotation, mis-annotations, etc.). These demonstrate that the PPV model is highly robust to misclassified training examples. Had these peptides not already been known, they would have constituted novel findings by the PPV model due to

their strong peptide-like patterns in the MS peptidomics data. To further underscore this point, we list a few additional in-silico validation examples in the revised manuscript (in the section “Model predictions and in-silico validation”).

Minor issues:

1. In the abstract (page 2), it is better to address the number of the peptides predicted in this study to indicate the size of this data resource, and also highlight these validated peptides with biological functions in this work.

We thank the reviewer for this suggestion, which we now have included such numbers in the abstract. We find that the high-level nature of the abstract is not suited for providing detailed breakdowns of prediction counts per tissue, threshold, etc. and instead simply say “hundreds” to indicate the approximate size of the resource with 251 unique peptides with a PPV score > 0.05 across all tissues. Updated detailed numbers for each tissue are provided in the main manuscript and supplementary tables.

2. In the line 7 of page 6, why “our data indeed reflects the in vivo situation”? Please explain.

In light of this comment, we deleted this particular sentence and rephrased the section. The point we were trying to make was that unsupervised dimensionality reduction (UMAP) and clustering on the peptide expression intensities (across all samples) automatically reproduces the grouping into tissue and strain, aka it emerges naturally from the data itself. We interpret this as a sign that the known biology (tissues, genetic background, and diet) is reflected in the peptidome which support that the observed peptidome is not simply a result of extensive and unspecific post-mortem degradation of normal house-hold proteins (common to all tissues). The other proteome vs peptidome comparison (in brain tissue) was meant to support this interpretation by showing that the peptide expression levels are poorly correlated with abundance of the proteins from which the peptides originate.

Since reviewer #1 also questioned the need for the protein-peptide correlation analysis, we revised Figure 2 so that now it only contains dimensionality reduction (Fig. 2a,b) and unsupervised clustering (Fig. 2c). The protein-peptide correlation analysis in brain has now been relegated to supplementary figure 5.

3. In the line 21 of page 6, please indicate the criterion for “perfect sequence matches”, such as Mascot score, sequential fragment ion number, etc.

We deleted this sentence, acknowledging that it can be misunderstood. We intended to say that we observed the exact same peptide sequence identical to the known annotated peptide. The analysis here has nothing to do with mass-spectrometry confidence/identification but is simply a matter of counting whether the exact amino acid sequence of each known peptide is present among the final set of observed peptides. We used this term perfect match to distinguish it from “partial matches”; cases where we only observe a sub-part of the sequence of a known peptide (e.g. a slightly shorter variant of the known peptide). We hope that the rephrased section will make this clearer to readers and thank the reviewer for alerting us to this.

4. In the line 15 of page 7, is it possible that the Manserin peptide is secreted, but is below the detection limit?

We agree with the reviewer that if the detection limit was lower in the Islet data set, then the missing identification of Manserin could be simply a result of it being expressed at a level below the detection threshold. The underlying reasons for not detecting it could therefore be either low general abundance, rapid extracellular degradation or the scenario that we point to: that Manserin was not secreted in these experiments because we failed to cover the conditions under which secretion occurs.

In the revised manuscript we have address this question further, based on the valid criticism by reviewer #2 and #1. We plotted in a new Supplementary Figure 6b, the peptide abundance (intensity) distributions of 1) all peptides in the Pancreas and Islet data sets, all peptides mapping to the Secretogranin-2 backbone for both data sets (Supplementary Fig. 6a, b), and 3) observations of Manserin itself across all samples from

Pancreas and Islet, respectively. What this shows is that the general peptide abundance is higher in the Islet data set than in whole pancreas tissue, both globally and specifically for the Secretogranin-2 backbone.

If Manserin follows the trend observed for other peptides in the same experiments, we should therefore expect it to be confidently identified (above the detection limit) in the Islet data, as it is in whole tissue data.

The peptide with the lowest abundance in the Islet data set is Transgelin-3 which has a log10 intensity of 5.85 and we can therefore assume this to be the lower detection limit in the Islet experiments. This is 10 times lower than the average expression level of Manserin in whole pancreas tissue and 100 times lower than the average expression level for other Secretogranin-2 peptides in Islets. We believe that these data make it highly unlikely that Manserin is present but below detection limit in the Islet supernatant.

This view is furthermore supported by Supplementary Figure 6a which shows concurrence between the Islet and pancreas data sets for all other peptides than Manserin (blue and green peptide clusters are aligned for all other peptides than Manserin). Lastly, we note that others have shown that Islet cells do produce Manserin under the right conditions (Tano, K. et al. Histochem. Cell Biol. 134, 53-57. 2010).

We therefore maintain the argument that the most likely explanation for not observing Manserin in the extracellular fluid of the Islet cells is that it is not secreted under the experimental conditions we use, and that this illustrates a particular strength of whole tissue peptidomics over ex-vivo cell models when it comes to coverage of peptides.

At the same time, we also acknowledge the argument from reviewer #1 that whole tissue peptidomics also comes with limitations, challenges and downsides (see comments on that later).

The new analysis was added as Supplementary Figure 6b and we also included a comment in this section of the manuscript to explain that whole tissue peptidomics has its main justification in a discovery-oriented setup.

5. In the line 5 of page 11, please show the MS/MS spectrum of this novel 22-amino acid peptide. To achieve a confident validation, it is better to fragment a chemically synthetic peptide and show the mirror MS/MS spectrum for a comparison.

It is important to achieve a confident validation especially of this important discovery highlighted from our model predictions. The alternatively spliced peptide was exclusively observed in pancreas but as independent observations in all 48 mice. In total, we measured the peptide 184 times in both a +2 and +3 charge state and with an average precursor mass error of 0.0725 ppm (0.000038 Da) and an average andromeda score of 226.199. We can confidently assign γ - and b -fragment ions to cover all peptide bonds in the sequence (see new Supplementary Figure 10a below).

As an additional control, we used the *Interactive Peptide Spectral Annotator* developed in Joshua Coon's lab (Brademan *et al.*, 2019, *Molecular & Cellular Proteomics* 18, S193–S201). The entirety of the γ -ion series can only be assigned if the amidation is present in the C-terminal including the γ_4 -ion (RENL_amidation). Theoretically the γ_4 -ion could be explained by a Glutamate (RENL) to Glutamine (RQNL) substitution in position 20 (and no C-term. amidation), however the b_{20} and b_{21} -ions containing the hypothetical E -> Q substitution do not exist (Supp. Fig. 10c).

We therefore conclude unequivocally that the novel 22-amino acid peptide is confidently identified including the C-terminal amidation. The spectral data has been added as Supplementary Figure 10.

6. In the line 10 of page 12, we can not exclude the possibility that the amidation is formed on the side chains of amino acids through glutamic acid to glutamine, and aspartic acid to asparagine, instead of C-terminal amidation.

We fully agree with the reviewer's principle point that a glutamic acid to glutamine substitution (or the equivalent D → N) could explain the amidation among the 542 predicted peptides (PPV score > 0.01) that contain at least one glutamic acid or aspartic acid residue, but logically this does not explain the amidation of the remaining 104 non-glycine amidated peptides which do not contain E or D.

Such substitutions could theoretically be explained by genetic variation among the mice (a GAA codon encoding a glutamic acid, could be changed to CAA encoding glutamine as example) but that is highly unlikely to be the explanation for hundreds of such observed peptides.

A post-translational enzymatic conversion of glutamate or aspartic acid to glutamine and asparagine in proteins has to our knowledge only been postulated in bacterial peptidoglycan formation (Figueiredo, T.A. *et al.* PLoS Pathog. 8, e1002508. 2012) and never suggested to occur in mammals. Furthermore, the Glutamine Synthetase and the corresponding Asparagine Synthetase, as well as Glutamate Decarboxylase, use free amino acids as substrate, and do not catalyze side-chain conversion post-translationally in formed poly-peptide chains.

Irrespectively of the potential mechanism, we addressed this question in the revised manuscript in Supplementary Figure 12 where we show manually curated MS/MS ion spectra of non-glycine amidated peptides containing an E or D (or both) for selected peptides with a high confidence ion score in the MS/MS spectra. Among these 8 examples, were unable to find a single example in which a E was converted to a Q (or D → N), validating that the amidation is positioned on the carboxy-terminus (see below).

However, in cases where the very last C-terminal amino acid is a glutamic acid or aspartic acid, it will not be possible to distinguish amidation of the C-terminal from amidation of the side chain. Only 108 such cases exist among our high-scoring peptides (PPV > 0.01) and we therefore consider the majority of our non-glycine amidated peptides to have the amidation group positioned on the C-terminus.

This result is independently supported by other studies that similar report non-glycine C-terminal amidated peptides (Ref: 20 & 21). Non-enzymatic C-terminal alpha-amidation through metal-catalyzed oxidative cleavage has been proposed as mechanism to explain these non-glycine amidated peptides (Ref: 21 and references therein). Our analysis has been added as Supplementary Figure 12.

7. In line 18 of page 12, the authors selected these peptide candidates for functional validation based on gene ontology annotations. Please explain the underlying mechanism. Is it possible to predict the potential functions of these predicted bioactive peptides on that?

We agree with the reviewer that this aspect was not well described or argued in the original manuscript.

As explained above, the PPV method can help to prioritize peptides based on their signal in MS data and sequence-based methods, like PeptideRanker, can provide additional information to support the case for a peptide being bioactive. What neither method can reliably do, however, is to infer the exact function of the peptide and that naturally raises the question of how to select the right assay to validate the function for a predicted peptide.

Other computational methods (also based on the peptide sequence) exist which predict more fine-grained functional categories, like “neuropeptide”, but the slightly increased level of granularity is still not sufficient to guide assay selections.

In this work, we approached this problem the other way around. At the company where many of the authors are or were employed, we had access to a range of established assays and animal models related to diabetes, obesity and metabolism. We therefore aimed to select peptides which originated from protein precursors where the gene/protein was associated in broad terms with such phenotypes. In the original manuscript, we mentioned only Gene Ontology annotations (to keep it simple) but one could also use literature mining, genotype-phenotype links (genetic variation, gene knockouts, overexpression, etc.), gene-disease links, etc. for the same purpose. This type of reasoning will probably work best for gene/proteins that only give rise to a single bioactive peptide (e.g. Neuropeptide Y, Peptide YY, Amylin, insulin, etc.) or to a group of peptides with related functions.

We have no systematic data to prove that this works, but none the less identified two peptides that showed activity in our assays and models. From some of the other reviewer comments, we realize that we had not done a good job at explaining the significance of those findings and have therefore added more experimental data and revised the manuscript to address the concerns raised over our experimental validation data (see other comments to reviewers).

In summary, we have addressed the present reviewer question by revising the section on “Experimental validation of predicted peptides in diabetes-related models” as well as better explaining what the PPV and PeptideRanker methods predict.

8. In line 17 of page 16, some mice are 15 weeks old and others are 26 weeks. Is there an age difference in the datasets?

The reviewer is correct that the two mouse models differ slightly in the age of the mice. For the genetically induced diabetic model (db/db vs db/+), the mice were 15 weeks old when the animals were terminated, and the tissues were harvested and snap frozen. The other model is based on feeding otherwise normal mice a low-fat versus high-fat diet to induce an obese phenotype in the latter group until the age of 26 weeks, after which the mice were terminated and tissue collected.

We cannot rule out a small effect of the age difference but in the unsupervised clustering of the samples based on peptide expression (Fig 2c) the effect of tissue origin is far greater than that of strain specificity, indicating that age has little, if any, influence on the peptidome.

More importantly, however, our experimental design was aiming at covering the diversity and heterogeneity of the entire peptidome space (to not miss peptides) and we therefore consider the differences in genetic background, strain, treatment, age, and tissue an asset rather than a limitation. If expression of certain peptides was indeed age-dependent, our design would potentially capture those peptides too.

Realizing that the original sentence in the Methods section may have been ambiguous, we added further details in the revised manuscript to improve clarity and thank the reviewer for bringing this to our attention.

9. In Figure 1b, the Mascot individually identified 26411 peptides, and MaxQuant is 30221. It is interesting to summarize the Mascot score and andromeda score on the three areas in the Venn diagram to see if the individually identified peptides has a low identification confidence.

We agree with the reviewer that the peptides identified uniquely in each search engine are of somewhat lower confidence. We have now made this analysis which completely supports the reviewer's comment, and have added this as a density plot in Supplementary figure 3i,j. Note, that we removed all peptides with a mascot score below 20 computationally, as detailed in the materials & methods section.

10. In Figure 2a and b, UMAP analysis were used for tissue and strain specificities, while PCA analysis was performed at protein level in Figure 2d. Why different analysis was used. How about the PCA analysis results on tissue and strain specificities?

We agree with the reviewer that the same method should be used for consistence and therefore now use UMAP instead of PCA for the protein-level analysis. Based on the input from Reviewer #1 (point number 2), we have moved the proteomics part from the main paper into supplementary Figure 5a. The clustering of samples by strain remains.

a

11. There is no x-axis label in the left and central panels of Figure 4a.

We thank the reviewer for finding this inadvertent omission. We have now added x-axis label to both panels.

12. In Figure 8b and c, there should have negative controls in the two experiments by injecting non-functional peptides.

We thank the reviewer for pointing this out and we agree that showing data for non-functional peptides is important to demonstrate the specificity of the read-out as an additional assay control.

To address this question of specificity, we now include additional experiments as Supplementary Figure 14a-d (see below). Peptides that were scored low by our PPV model were chosen as non-functional peptide controls. Their PPV score are approximate 10x fold lower than the lower threshold (PPV score > 0.01) we employ for considering a peptide potentially interesting.

Protein	Peptide pos.	PPV score
CHGA	309-403	PPV: 0.0035
PNLIP	169-181	PPV: 0.0018
CHGA	374-390	PPV: 0.0019
MYDGF	124-166	PPV: 0.0005
COPA	756-793	PPV: 0.0005
CASQ1	35-59	PPV: 0.0036

The first experiment is an acute blood glucose measurement with three additional peptides made by array synthesis, some of which are from the granin family, from which our NHPD-50 peptide also originates. None of these low scoring peptides show a significant blood glucose lowering effect (Fig 14a), in contrast to the novel NHPD-50 peptide from Secretogranin-1 (SCG1: 386-435, Fig 8a), the exendin-4 peptide (positive model control) or other positive controls such as Islet amyloid polypeptide (IAPP: 56-74) or Peptide tyrosine tyrosine (PYY: 3-36) (see Suppl. Fig. 13a,b). In conclusion, injecting randomly selected peptides at these doses will not produce a functional read-out in this animal model whereas those peptides known to affect blood glucose do.

Next, we chose a longer peptide derived from Myeloid-derived growth factor (MYDGF: 124-166) which – similar to our NHPD-50 peptide – is required to be produced in *E. coli* (due to length limitation in chemical array synthesis). This time, the acute blood glucose experiment was done with a much higher dose 2600 nmol/kg (instead of 1000 nmol/kg normally used). Even at this high dose, injecting the low-scoring non-functional MYDGF peptide had no effect on blood-glucose (Supp. Fig. 14b).

In the original manuscript, we reported a follow-up confirmatory study with NHPD-50 (Fig 8c) demonstrating that a sustained blood glucose lowering effect could be achieved over 8 hours by dosing the peptide multiple times. Following the suggestion by the reviewer, we included a similar study using the MYDGF: 126-166 peptide at comparable doses (Supp. Fig. 14c). It shows no change in blood-glucose and thus supports the conclusion that the effect observed with NHPD-50 is both reproducible and specific to this peptide (aka not a result which can be achieved by high sustained dosing of any peptide).

To furthermore demonstrate that the increase in plasma insulin observed with NHPD-50 dosing (SCG1: 386-435) is indeed a peptide-specific response, we now include experimental data from two other peptides; COPA (756-793) and CASQ1 (35-59) as Supplementary Fig. 14d. Both of these peptides are predicted to be non-functional or non-peptide-like by our PPV method (score 0.000468 and 0.003596, respectively). Due to formulation and solubility constraints, it was only possible to dose the COPA and CASQ1 peptides at 354 nmol/kg and 402 nmol/kg, respectively (compared to 2658 nmol/kg for NHPD-50). As expected, the COPA and CASQ1 peptides did not elevate plasma insulin (Supp. Fig. 14d) or increase the C-peptide concentration in plasma (not shown). Contrasting this with Figure 8c of the main paper support the original conclusion of

the insulin secretion being driven by the NHPD-50 itself and not an effect which can be achieved by injecting random peptides at high doses.

Collectively, the new data demonstrates that other peptides with a different amino acid composition are incapable of mediating any of the effects seen with NHPD-50, such as the acute (Fig 8a) and sustained (Fig 8b) blood glucose reduction or the significant increasing plasma insulin concentrations (Fig 8c).

We hope that the reviewers will agree that these additions to the manuscript support the specificity of our models and the conclusion that the demonstrated bioactivity of NHPD-50 is not simply an artifact resulting from high dosing.

Reviewer #1 (Remarks to the Author):

This manuscript describes the generation of a database of peptides identified by mass spectrometry from a range of mouse tissues, together with the development of a prediction programme, potentially able to identify bioactive peptides from mass spectrometry data (the needle in the haystack). The programme was trained against peptidomics data from a variety of tissues which included known bioactive peptides, and then used against a much larger dataset. It was ultimately capable of picking up the peptides against which it was trained, as well as others that had either been mislabelled, or that were not in the training dataset.

Overall, the programme will certainly be of value to researchers in the field if made freely available, as the authors are correct in stating that it is laborious, and near impossible, to identify potentially bioactive peptides by manually sifting through mass spec peptidomic databases that contain degradation fragments and non-active products of proteolytic cleavage reactions.

We thank the reviewer for appreciating the model and code which is freely available for the scientific community to use and expand upon. We are also pleased by the reviewer's remark on the importance of new approaches for identifying potential real bioactive peptides from large-scale peptidomics studies.

I am less convinced, however, that the authors were successful in identifying new bioactive peptides using this approach. They describe a few peptides identified by the algorithm that were taken through to in vivo and in vitro studies, but these were only effective at very high concentrations in a limited number of experiments, and the in vivo findings could not be backed up by in vitro mechanism of action studies.

We fully acknowledge that we do not in this work provide solid, conclusive evidence for the exact function and mode-of-action of the two example peptides which show bioactivity in our assays and models.

But, since the main focus of the manuscript is to share a new method which can help to identify novel candidate peptides from MS-based peptidomics data, we hope that the reviewer will appreciate the combination of all the performance metrics, enrichment analysis and in-silico validation examples which demonstrate that the PPV method can indeed find bioactive peptides that it was never trained on (in fact, many of these were labelled as negatives during training), as well as peptides which share the same characteristics (cleavage sites, secretion, high PeptideRanker scores, etc.) as already known ones.

In addition to this, we still believe that our experimental data support the claim that these particular peptides do exhibit bioactivity in some form, even if the effect may be indirect and the mechanism unknown.

The reviewer is correct that peptides are being administered at high doses in the in-vivo studies, but this is often required to overcome renal clearance as well as rapid peptide degradation in the bloodstream, unless the peptide is engineered for stability. But we can testify – as employees of one of the world-leading companies within discovery, development and production of peptide-based therapeutics – that observing any effect at all in these assays is a very rare event which can not simply be achieved by administering high doses of randomly selected peptides. Even the most potent known peptides result in what may appear as modest responses. The reviewer's comments have made it clear to us that this aspect was poorly explained in the original manuscript and we have consequently added additional experimental data and made changes to the manuscript better get this message across to readers.

We have added additional data to show that other peptides from the same protein family (other granin-peptides from our MS data that were scored very low by the PPV method) do not elicit any response, even when dosed at similar levels. These peptides now act as a kind of negative controls in the manuscript as

Supplementary Fig. 14a

depicted in Supplementary Fig. 14a-d (see also comment above).

In addition, we have added data on GLP1, a known blood-glucose lowering peptide that induces insulin secretion, to illustrate that even with high dosing of this very potent endogenous peptide, the effect is significant but transient. This is due to the short half-life of most native peptides. It is only when administrating unnaturally stable peptides, like the marketed drug product Exendin-4 (our positive control), that a major sustained effect on blood glucose is observed.

Supplementary Fig. 13b

The fact that the NHPD-50 peptide from Secretogranin-1 elicits a significant response (Fig 8a) is therefore highly unlikely to happen by chance, particularly when the finding can be replicated in a multiple-dosing follow-up study (Fig 8c). We find this to be very strong evidence of some bioactivity of the predicted peptide and therefore believe that the results constitute experimental confirmation of the validity and utility of our approach.

We fully acknowledge that based on the in-vitro data for this peptide, the in-vivo effect on blood glucose may be indirect, rather than direct, and have tried to make this point clearer in the manuscript (section “Experimental validation of predicted peptides in diabetes-related models”).

We appreciate the reviewers concern that none of the two examples are fully validated functionally, but we feel the significance in the manuscript lies primarily in the model’s ability to highlight peptides for further investigation.

Specific points:

1. The authors have referenced other bioactive peptide prediction software resources on page 4 – for example peptide ranker by Mooney et al. How do the peptides that were selected for administration rank on these other peptide prediction packages compared to PPV?

We thank the reviewer for the good idea and have added a direct head-to-head comparison to PeptideRanker in our new model performance evaluation section (Fig. 4c, Supplementary Fig. 7a). The PeptideRanker code is not publicly available, but Catherin Mooney kindly shared it with us, so that we could run all our observed peptide sequences through it to get a prediction for each one.

Fig. 4c

As mentioned above, the comparative analysis demonstrates that our PPV method (which now only uses the patterns in MS data as input) clearly outperforms PeptideRanker (which takes only the amino acid sequence of the peptide as input) when it comes to identifying known annotated peptides. Note that this comparison actually gives an advantage to PeptideRanker over PPV. For our method, the prediction scores are computed only from the models that were not trained on the peptide in question. In contrast, PeptideRanker (or at least part of it) was most likely trained on the majority of the known peptides and may thus for many of them be predicting on its own positive training examples when we run all the known peptide sequences through the tool. Despite any bias being in favor of PeptideRanker, PPV is far better. This demonstrates that there is more signal in the MS data than in the sequence composition of the peptides.

Since the methods are now completely orthogonal (use entirely different input data for their predictions), we made a new figure (Fig. 5e, see next page) where we plot the PeptideRanker scores for peptides which are scored high by PPV to highlight new potential peptides which score high on both methods.

Our NHPD-50 peptide from secretogranin has a PeptideRanker score (0.336) on par with known bioactive peptides like NERP-1 and Neuropeptide K, and the novel D-peptide from insulin scores even higher (0.495) at the level of Manserin, Neurotensin and Neurokinin A. The ATPR-15 peptide from Tpi1 also really stands out with a score of 0.844 putting it at the level of VIP, Neuromedin-C, Secretin, Orexin-B and Pancreatic hormone. The assembled Gelsolin peptide scored the lowest among our examples (0.124).

Lastly, PeptideRanker scores are now included for all predicted peptides as a new column in Supplementary table 5, which we hope will aid the reader in sorting the peptide data according to one or both prediction schemes.

Fig. 5e

2. The proteomics part of the paper should arguably be cut, as it isn't clear how it adds to the manuscript. The authors say that the proteomics data back up the strain separation for figure 2B compared with figure 2D, however their explanation for including it to confirm the peptides aren't derived from degradation of large proteins is not only confusing, but also a very weak argument. I don't see the need for including this proteomics data set in a paper discussing the identification of bioactive peptides using mass spectrometry.

We see the reviewers point here and agree that the proteomics part contributed to making this section too long and overly complex to digest. We therefore relegated it to the supplement (Supplementary Fig 5a,b) and reduced Fig 2 in the main manuscript to only include the peptidome analysis (panels a-c). We also rephrased parts of the section to improve readability based on other reviewer feedback.

The reason for not entirely removing the additional experiment (the one comparing the proteome to the peptidome) is to allow interested readers to see the relatively modest correlation between the traditional protein abundance estimates (proteomics based on tryptic digest) and estimates of protein abundance based on transferring abundance values for the observed peptides in a peptidomics experiment.

If the peptides observed were merely a result of degradation, one should expect the abundance-level of each protein to largely dictate the abundance of peptides/fragments from that protein. This would result in

a perfect correlation in Supplementary Figure 5b which is far from what we observed. In fact, the correlation coefficient is only 0.20 indicating that the abundance of peptides from a given protein is a poor predictor of the protein abundance, unlike what is seen in proteomics experiments. We therefore believe this supports the notion that whole tissue peptidomics data is not just a result of random degradation of household proteins.

3. Figure 3b is particularly difficult to follow and interpret. Whilst this is a potential way to show degradation, using just peptide spectral matches doesn't necessarily show the whole picture. Likely the authors have already attempted different ways of displaying these results. However, one alternative would be to compare the peak area or intensity of each matched peptide relative to the parent peptide. This will indicate the relative levels of degradation from each method, as well as which peptides might be generated at sufficient quantities to have bioactive potential when diluted into the bloodstream.

If intensity data aren't available from the other data sets, another way of showing the data would be to plot the number of peptides identified for each protein as a function of the parent protein molecular weight. The majority of bioactive peptides come from low molecular weight precursor proteins (bar some of the non-canonically cleaved peptides) therefore this approach would also highlight the challenges in differentiating peptides that are generated from unwanted protein degradation and those that are truly real bioactive peptides.

We agree completely with the reviewer that our initial plot was not the most intuitive and thank the reviewer for the reflections and suggestions for improvement.

As the reviewer correctly points out, the key limitation in making the ideal comparison is that the intensities (abundance estimates) of individual peptides are available to us from our own study but not from the other studies we wish to compare to. We are therefore limited to comparisons based on peptide counts (present vs not present).

Building on the ideas of the reviewer, the updated Figure 3b is now based on taking all the known annotated peptides that were observed in each of the four studies and then plotting the number of shorter fragments observed (y-axis) as a function of peptide length (x-axis). Here, the count of shorter fragments is used as a proxy of degradation assuming that all shorter fragments represent inactive degradation products (although they may in some rare cases represent other shorter variants that act as bioactive peptides in their own right).

The analysis confirms that other recent peptidomics studies also suffer from degradation to various extent across different cell systems, cell enrichment procedures or tissue peptidome methodologies. Our Ileum data looks slightly more degraded in comparison to some of the enteroendocrine cell systems and the gastrointestinal tract data but it is difficult to determine if this could – at least partly – be due to an increased sensitivity in our study which includes data from 48 mouse Ileum samples. We generally found that adding more samples led to detection of more unique peptides both within and across tissues (Fig 1b).

As would be expected statistically, all four studies show a tendency for longer peptides to have more shorter fragments (be more degraded).

In conclusion, all peptidomics studies display degradation to some extent even though specific experimental procedures are used in all studies to prevent post-mortem degradation. The degradation is therefore most likely a real in-vivo phenomenon which underscores the need for a computational approach deal with this complexity. Our PPV model is one such method that turns the problem (degradation) into a solution by learning from the topology of the degradation patterns which are otherwise considered noise.

4. A small subset of pancreatic sample data files were downloaded and put through a peptidomics search by this reviewer. Surprisingly, there were no A or B chain matches against insulin in the data. I couldn't see reduction and alkylation being discussed in the method section (information on page 18 doesn't include carbamidomethylation). If reduction and alkylation isn't performed, then a large subsection of bioactive peptides could be missed. The supplemental data set (supplementary figure 2) shows data where peptides were reduced and alkylated to check the recovery of the MWCO devices – but it wasn't used in the main experimental dataset? Please discuss your reasoning for not performing reduction and alkylation on the main dataset.

It is correct that we did not use reduction and alkylation during this peptidomics study, and we agree with the reviewer that this inevitably results in a subset of peptides being missed in our analysis.

Disulfide-bridges are structurally and functionally important for certain bioactive peptides, especially in antimicrobial peptides such as defensins or in peptide toxins from venomous animals, but importantly also - as suggested by the reviewer - within some peptide hormones, most prominently insulin and the Cart peptide. Such peptides are "invisible" in our study. From our gold standard list of known annotated peptides, 98 cysteines containing ones were not present in our data (Supplemental Table. 3). Of these 98, 35 are defensins which are primarily produced in epithelial cells, leukocytes, and in the intestine.

The reasons for not using reduction and alkylation are three-fold.

Firstly, we did initially try reduction and alkylation, but observed a drop in identification rate (Supp. Fig. 2a) with a very limited gain in cysteine containing peptides. We therefore prioritized the coverage of other categories of peptides at the expense of missing a small group of cysteine containing peptides.

Secondly, we knew upfront that testing for bioactivity would be performed within a defined biological screening setup optimized for finding peptides that can lower blood glucose acutely when injected into diabetic mice and/or induce a response in metabolic in-vitro assays. To screen cysteine containing peptides in these models, we would first need to disentangle non-trivial inter-molecular disulfide-bridge configurations, as seen within insulin, or - if they occur intra-molecularly - in closely spaced nested patterns. This would require more advanced methodologies such as in-source reduction during the ionization process

(Cramer, C.N. *et al. Anal. Chem.* 89: 5949-5957. 2017) that would be difficult to scale to the number of samples we aimed to analyse.

Thirdly, the production of the candidate peptides was performed using an automated array-based Fmoc solid phase synthesis setup in which we could not properly control the cysteine-bridge configurations and hence not ensure that cysteine-containing peptides were synthesized correctly (misfolding, racemic mixtures, etc.).

In summary, we therefore put lower priority on identifying cysteine containing peptides in this study due to the difficulty in producing and testing them.

We added this specifically to the “Experimental setup” section of the main manuscript and also touch upon it in the revised discussion section at the end of the manuscript.

5. The pancreas peptidomics data also had a low signal for the glucagon peptide in a fair number of the samples. I would have expected a much higher signal in the pancreas for glucagon, as this is one of the main peptide products from the pancreas. Coincidentally, there was a significant amount of degradation of the glucagon peptide as well as the other peptides from the proglucagon gene. Although this doesn't reduce the validity of the algorithm for identifying peptides, it does highlight the issue the need to optimise tissue processing to preserve native peptides, which should be discussed.

We thank the reviewer for these reflections.

First, we would like to point out that we followed best experimental practice (References 27 and 28) for tissue peptidomics to minimize post-mortem degradation. We snap froze all tissues immediately after removal. Tissues were then taken directly from a frozen state and heated rapidly to 95C in a Denator heat stabilizer to inactivate protease and peptidase activity before peptide extraction. We also conducted an additional test experiment (Supplementary Fig 2c) to show that perfusion with a protease-inhibitor containing saline solution (before tissue removal) resulted in the same peptide length distribution, indicating that the snap freezing procedure does not induce degradation of the peptides.

This of course does not completely eliminate the possibility of some post-mortem degradation but as shown in the updated Figure 3b, degradation is also observed in other studies although these too employ measures to prevent degradation.

It is inherently difficult – if not impossible – to fully determine to which degree the degradation we (and others) observe reflects the true in-vivo situation in the tissues studied versus post-mortem degradation resulting from the experimental procedure, but we do not think that the latter is the most likely given the above arguments.

The reviewer specifically downloaded samples from pancreas to check the abundance of pre-proglucagon-derived peptides in pancreas, expecting to find glucagon to be more abundant than its degradation products and other peptides. We acknowledge and confirm that this seems to indeed be the case for this particular peptide, although it is detected across all strains and conditions. This could be the result of the short half-life of the active peptide (4-7 minutes in human plasma according to Rix *et al.*, 2019, www.ncbi.nlm.nih.gov/books/NBK279127/) and/or influenced by the fact that glucagon also exists in a phosphorylated form (on serine 54 according to Huttlin, *et al. Cell.* 143:1174-1189, 2010). To investigate this, we re-searched our pancreatic samples from the diabetic strain background, which has the highest

degree of glucagon, but adding phosphorylation on S/T as a variable modification. This confirmed the existence of the phosphorylated variant which is otherwise “invisible” in our data set (data not shown). The abundance estimate of glucagon might therefore not be fully representative of the *in-vivo* situation. As pointed out elsewhere, expanding the theoretical PTM space is, however, not computationally tractable in a global analysis of the entire data set.

As pointed out by the reviewer, the degradation observed for the glucagon precursor protein – whether *in-vivo* based or experimentally induced – illustrates the power of the algorithm in overcoming the complexity of the peptide processing. In pancreas, the highest scoring peptide is Glicentin-related polypeptide (GRPP) known to be produced specifically in pancreas whereas GLP1(7-36)am is the top ranked peptide identified in the intestine (Ileum) which is known to secrete this peptide in response to food intake.

This suggest that our data and algorithm reflect the known processing of pro-glucagon in these two tissues, although lower scoring degraded peptides are also present.

We have updated the degradation analysis (Fig 3b) and rewritten parts of the manuscript to better explain to readers how the PPV method works and what its benefits and limitations are, as well as revised the discussion to highlight limitations of the experimental methodology, including the possibility of missing peptides with distinct post-translational modifications (like the phosphorylated Glucagon variant).

6. Can the authors include figures showing spectral data and annotated product ion spectra for the novel insulin peptides – this would give more confidence in the identification of the splice variants that also contain the C-terminal amidation modification.

We thank the reviewer for this suggestion which we have implemented as a new supplementary figure 10.

In the exemplary MS/MS spectrum (see below), the observed fragment ions have an average mass error of only 1.608 ppm and cover the entire peptide sequence illustrating the high confidence of the identification. This novel insulin peptide was observed independently in all 48 pancreatic samples and is the highest scoring PPV prediction in pancreas (Supplementary Table 5). Additionally, we now also report the PeptideRanker prediction score which supports a potential bioactive role for this alternatively spliced peptide that – to our knowledge -have never been described before.

The reading frame established by the alternative splicing event ends with a Glycine before the stop codon (Fig. 6b), but the observed peptide ends with a C-terminal amidation group on Leucine confirmed from the y-ion series. This strongly implies that the amidation modification is mediated by the PAM enzyme which converts terminal Glycines to the amidation. The new spectral data has been added as Supplementary Figure 10a. Panel b of the same figure furthermore confirms the C-terminal position of the amidation.

Supplementary Fig. 10a

7. The authors describe the ability to detect Manserin in the whole pancreatic tissue (page 7), but not in islet supernatants, which they use as an argument for analysing tissue peptidomics to identify peptides for which the secretory conditions are unknown. However, this approach will also identify a large number of peptides that are never secreted, and are effectively false positives. The lack of detection of manserin in islet supernatants is likely due to the sensitivity and recovery of the secretory peptidomics being sub-optimal. This limitation should be discussed, together with some discussion of the authors' opinion about the likelihood that individual peptides identified by the algorithm, but deriving from large cytosolic proteins, will turn out to be genuine bioactive peptides rather than false positives.

Addressing the last point first, we agree completely that tissue peptidomics will be hampered by a large number of peptide sequences formed as natural turnover of cytosolic proteins, and unlikely in most cases to possess any ingrained activity. We now discuss this in the final section of the manuscript ("Discussion") along with other limitations and pitfalls of our data and methodology, taking into account the good points of reviewer #2 pertaining to how to best make use of the prediction for subsequent experimental follow-up.

Although most cytosolic peptides are probably non-active, there is growing evidence that cytosolic derived peptides can be translocated across the membrane and secreted through non-canonical pathways (i.e. *not* through the ER/Golgi/secretory pathway) via alternative mechanisms such as; micro-vesicles, secretory exosomes, or by proteasome association with the membrane (Nickel W, Rabouille C. *Nat. Rev. Mol. Cell Biol.* 2009, 10: 148-155 & Ramachandran KV, Margois, SS. *Nat. Struct. Mol. Biol.* 2017, 24: 419-430). Furthermore, some of these non-canonical and cytosolic derived peptides have been found to be bioactive as summarized in Supplementary Figure 1 and highlighted in the revised manuscript.

We still predict on all observed peptides in the global data set but offer the reader the possibility to filter away cytosolic peptides in the final result tables. We do, however, wish to highlight in the manuscript an example of a novel cytosolic peptide which the PPV method singles out in the vast sea of other cytosolic peptides. This peptide (APTR-15) is observed in multiple tissues and in the revised manuscript further supported by a very high PeptideRanker score, on par with or better than well characterized peptides like pancreatic hormone, Orexin-B, Gastrin-releasing peptide and VIP (Fig. 5e, Supplementary Table 5). For this

peptide, we also provide structural support for the potential cleavage site to be in an exposed loop of the enzymes' 3D structure (supplementary Figure 11).

The question of Manserin detection (and other intracellularly stored peptides) was also brought up by reviewer #2 and we refer to our comments above (reviewer #2, point 4) and the newly added analysis (Supplementary Fig. 6b).

We have expanded the discussion of the limitations and challenges of whole-tissue peptidomics in our revision but also note that sampling from cell systems confer a risk of missing peptides because they do not capturing the complexity of an entire tissue and require establishment of the secretion conditions for each cell system.

8. With the atypical C-terminal amidation findings for peptides, how certain are the authors that these are real and not instrumental artifacts? For larger peptides, the precursor ion that is selected for fragmentation can sometimes be erroneous – giving an apparent ~1 Da mass reduction. If the subsequent fragmentation contains mainly b-ions – which is quite common in peptidomics, then this might also give good peptide sequence matches with good mass accuracy, and therefore an incorrect C-terminal amidation assignment. Have these atypical modifications been confirmed by manual curation of the precursor ion C13 cluster and the respective product ion spectra?

We agree with the reviewer that it could be a cause for concern if amidation assignment is not supported by y-ion series in the fragment-ion spectra. In this study, we employed a QE-HF orbitrap instrument with HCD fragmentation which provides high mass accuracy both at the precursor and the MS/MS level. In the MaxQuant search engine peaks are detected in each MS scan by fitting a gaussian peak shape to the raw data points and subsequently assemble them into a 3D-peak over the m/z retention time plane (Cox, J. & Mann, M. *Nat. Biotechnol.* 2008, 26: 1367-1372). The high precision 3D peak includes the isotope distribution. From this information the software takes into account which isotope was picked for fragmentation, and does not wrongly assign an incorrect precursor mass, such as a C13 picked isotope value to a C12 mass.

This is exemplified in the figure below. Here, the C13 isotope of the non-glycine amidated peptide (m/z = 1317.1011) was picked for fragmentation. The monoisotopic precursor of the peptide was subsequently correctly annotated as m/z = 1316.5981 demonstrating that the software automatically takes into account which isotope was picked for fragmentation (C12 vs. C13). Independent on which precursor isotope was selected for fragmentation, the C-terminal amidation can unambiguously be assigned based on the y-ion series.

We agree with the reviewer that it is important to confirm the atypical amidation, and we have now curated non-glycine amidated product ion spectra manually (see also reviewer #2, point.6) and added this as Supplementary Figure 12.

9. The experiments in suppl fig 6 claim to validate the importance of C-terminal amidation for neuromedin C. However, the non-amidated version was only tested at a lower concentration in vivo, and was not tested in the insulin secretion experiments. To conclude that amidation is important, the amidated and non-amidated peptides need to be compared at the same concentrations and doses. Notably, stimulation of insulin secretion in vitro was only observed at very high concentrations (10 and 20 microM), arguing against this targeting a specific high affinity receptor.

We thank the reviewer for these comments. We agree that we lacked comparable in-vitro data on both variants and have now added the missing data to enable the comparison (as supplementary Fig 8c).

These data clearly demonstrate that a significant effect on insulin secretion is only obtained with very high doses of the non-amidated variant of Neuromedin-C (NMC) whereas the amidated variant significantly induces secretion at all concentrations tested.

When it comes to the in-vivo data, we realize that the figure may not have been sufficiently clear in communicating the results. We did inject both the non-amidated and the amidated NMC at the same

concentrations (1000 nmol/kg) in comparable in-vivo experiments using 7 mice. We have now updated supplementary Figure 8 (panels a and b) and the legend to make this and the appropriate comparison clearer.

The original version of the manuscript also included an additional follow-up study in which we dosed the mouse and human amidated variants at even higher concentrations to compare the two. Realizing that this detracts attention for the main message (the importance of amidation), we have removed this panel in the updated supplementary figure 8 to avoid misunderstandings.

The new supplementary Figure 8 is depicted below.

REVIEWERS' COMMENTS

Reviewer #1 (Remarks to the Author):

This is an extensive rebuttal, and the resultant changes have improved the manuscript. It is a piece of work that will be of value to the research community.

Reviewer #2 (Remarks to the Author):

The authors have tried to answer the reviewers' questions, but it seems that there still have some issues. The new prediction tool fails to classify the bioactive peptides. In other word, the software tool can predict bioactive peptides, but it cannot predict bioactivity. This limits its usefulness. In addition, the authors did not highlight the modifications in the revised manuscript, which cause difficulty in review.

1. Regarding the response to major issue 1, is it possible to predict the peptides' bioactivity from the known annotated peptides? This will be useful for the following functional validation experiment.
2. Regarding the response to minor issue 6, several MS/MS spectra for the annotation of C-terminal amidation is not confident, such as the spectra in Panels c and d. Substitution of E to Q could happens, instead of C-terminal amidation. Is that possible to set up a threshold for localization possibility score (Mascot) to improve the identifications?

Reviewer #3 (Remarks to the Author):

All of my comments have been addressed in a satisfactory manner. Therefore, I recommend this paper for publication.

Rebuttal letter

Reviewer #1 (Remarks to the Author):

This is an extensive rebuttal, and the resultant changes have improved the manuscript. It is a piece of work that will be of value to the research community.

We thank the reviewer for the kind words and for the many good suggestions that have helped us improve the manuscript.

Reviewer #3 (Remarks to the Author):

All of my comments have been addressed in a satisfactory manner. Therefore, I recommend this paper for publication.

We thank the reviewer for the comments and suggestions which have helped us improve our manuscript.

Reviewer #2 (Remarks to the Author):

The authors have tried to answer the reviewers' questions, but it seems that there still have some issues. The new prediction tool fails to classify the bioactive peptides. In other word, the software tool can predict bioactive peptides, but it cannot predict bioactivity. This limits its usefulness.

We agree with the reviewer that our method does not predict the exact bioactivity of a peptide and we also acknowledge that a method capable of pointing out the mode of action and/or biological role of a peptide directly from the data would be highly useful. We have added to the discussion in the manuscript to expand on this limitation.

In addition, the authors did not highlight the modifications in the revised manuscript, which cause difficulty in review.

We are truly sorry that the changes were not visible to the reviewer and fully understand the frustration. We did in fact use track-changes to highlight all changes in the manuscript and note that none of the other reviewers appear to have experienced difficulties in review. We do not know if the changes may have been lost after we submitted the revised manuscript, for instance in conversion to PDF.

1. Regarding the response to major issue 1, is it possible to predict the peptides' bioactivity from the known annotated peptides? This will be useful for the following functional validation experiment.

We fully acknowledge the challenge of functionally validating peptides in the absence of detailed predictions of their mode of action or biological role. We now emphasize this aspect in the revised manuscript, both in the section on experimental validation and in the discussion. We do, however, also wish to point out that detailed functional predictions have not been the aim of our study and we doubt that such detailed prediction could realistically be derived from the MS data, given the nature of the data and the relatively small number of known peptides to train on.

2. Regarding the response to minor issue 6, several MS/MS spectra for the annotation of C-terminal amidation is not confident, such as the spectra in Panels c and d.

The peptides in Supplementary Figure 12 are all examples of peptides that contain a Glutamic acid (E) or Aspartic acid (D) and for which our analysis has assigned an amidation of a non-Glycine terminal residue. The purpose of the Figure is to exclude the possibility of amidation anywhere else but the c-terminal for these examples. We show that high-confidence y-ions with high mass accuracy exists that unambiguously assigns the amidation (mass difference of 0.98 Dalton) at the C-terminal and rules out amidation elsewhere in the peptide chain.

Specifically for Figure 12c (mentioned by the reviewer), we observe the y4 ion which could theoretically be explained by either EDPV(am) or alternatively by ED(Am)PV, as the reviewer suggests. The latter option can, however, be ruled out based on the y2 ion. It is observed with high mass accuracy (albeit low intensity) and can only be explained by PV-am (PV-dipeptide amidated on V). This logically rules out the possibility of an amidation on the Aspartic acid (D) in position 7 of the peptide. Similarly, in Figure 12d, we observe both y1 and y2 ions (again with high mass accuracy) corresponding to P-am and VP-am, respectively. These observations rule out the possibility that the y9 ion could be explained by amidation on either E or D, leaving C-terminal amidation as the only possibility to explain the observed mass differences. The same type of arguments applies to the additional six examples shown in Supplementary Figure 12.

Based on this, we do not agree with the reviewer that these spectra are not confident and that the amidation could be anywhere else than on the C-terminal for these peptides.

Substitution of E to Q could happen, instead of C-terminal amidation.

As we write in the manuscript, there are examples where the MS/MS fragment pattern is of too low quality to unambiguously assign the amidation to be C-terminal. In these cases, it is theoretically possible that an enzymatic (or non-enzymatic) mechanism could convert side chain carboxylic acids on D and E to Q and N, respectively. This, however, has not - to our knowledge - been demonstrated to actually occur in already formed polypeptide chains, and we are not aware of any endogenous enzymes capable of performing such conversions. Non-enzymatic chemical conversion would require very high temperatures (essentially boiling) and very high concentrations of ammonia (that does not exist in-vivo).

We discuss these theoretical possibilities in the manuscript but also openly declare that we consider them highly unlikely to explain the apparent C-terminal amidations that have also been observed by several other groups (Ref: 20 & 21). Non-enzymatic C-terminal alpha-amidation through metal-catalyzed oxidative cleavage has been proposed as mechanism to explain these non-glycine amidated peptides (Ref: 21 and references therein).

Is that possible to set up a threshold for localization possibility score (Mascot) to improve the identifications?

In the current search engines, it is not possible to apply a threshold for C-terminal amidation score, although this approach is used for phosphorylation (work from Bernhard Kuster: <https://pubmed.ncbi.nlm.nih.gov/21057138/>).

In conclusion, we have added extra sections within the manuscript and in the discussion that explain the limitations of the PPV algorithm. We wish to thank the reviewer for the substantial contribution that has improved our manuscript considerably.